# p53 regulates DREAM complex-mediated repression in a p21-independent manner

Ritu Agrawal (ID)[1] & Sagar Sengupta (ID)[1,2✉]

## Abstract

The DREAM repressor complex regulates genes involved in the cell cycle and DNA repair, vital for maintaining genome stability. Although it mediates p53-driven repression through the canonical p53-p21-Rb axis, the potential for p53 to directly regulate DREAM targets independently of its transcriptional activity has not been explored. Here, we demonstrate that in asynchronously growing cells, p53 loss leads to greater de-repression of DREAM targets compared to p21 loss alone. Both wild-type and transactivation-deficient p53 mutants are capable of repressing DREAM targets, suggesting a transactivation-independent "non-canonical" repression mechanism. These p53 variants bind p130/p107, irrespective of their phosphorylation status, while cancer-associated p53 mutants disrupt DREAM complex function by sequestering E2F4. Re-ChIP analysis shows co-recruitment of p53 and E2F4 to known and newly identified DREAM target promoters, indicating direct repression of these targets by p53. These findings reveal a novel, transactivation-independent mechanism of p53-mediated repression, expanding our understanding of p53's tumor-suppressive functions and suggesting DREAM complex targeting as potential future avenues in cancer therapy.

**Keywords** DREAM Complex; p21; p53; Transcriptional Repression; Transactivation-independent p53
**Subject Categories** Chromatin, Transcription & Genomics; Signal Transduction

## Introduction

Tumor suppressor p53 is essential for maintaining genomic stability. Both the complete loss or inactivation of p53 by mutations lead to cancer development (Joerger and Fersht, 2016; Kastenhuber and Lowe, 2017). p53 gene is mutated in approximately 60% cases of colorectal cancer (CRC). The most common p53 missense mutations in CRC (R175, R248, R273, R282) account for approximately 37% of the mutation load in this gene (Hassin et al, 2022). Wild type (WT) p53 is a well-known transcriptional activator with well-defined mechanisms (Kastenhuber and Lowe,

2017). However, the role of p53 as a repressor has been re-evaluated in the recent past. p53 has been known to directly repress genes by binding to its response elements and recruiting corepressors or by blocking transactivators from accessing their binding sites (Brady and Attardi, 2010; Ho and Benchimol, 2003). However, more recently, a meta-analysis of data published in multiple studies has led to the conclusion that p53 represses transcription indirectly by activation of the p53-p21-DREAM complex pathway (Sullivan et al, 2018b; Fischer et al, 2014). Apart from the role of p53 as a transcriptional activator and repressor, p53 also has transactivation-independent functions, which also play defined roles in maintaining genomic stability (Raj and Attardi, 2017). For example, the role of p53 in regulating multiple DNA repair processes and homologous recombination in a transactivation-independent manner has also been reported (Bertrand et al, 2004; Sengupta and Harris, 2005).

The DREAM complex (dimerization partner, Rentinoblastoma-like, E2F, and multi-vulval class B) is a transcriptional repressor complex known to regulate genes involved in cell cycle progression and proliferation, as well as DNA replication and repair processes (Sadasivam and DeCaprio, 2013; Engeland, 2018; Bujarrabal-Dueso et al, 2023). The DREAM complex becomes functional when p53 is activated, which leads to the induction of the cell cycle kinase inhibitor p21 expression. When p21 inhibits Cyclin Dependent Kinase (CDK) activity, retinoblastoma (RB) family members p107 and p130 become hypo-phosphorylated, resulting in their binding to E2F4 and forming the DREAM complex, which in turn suppresses gene expression of its targets by binding to their promoters (Engeland, 2022). It is accepted that the DREAM complex regulates genes in a cell cycle-dependent manner, repressing genes only during the G0 phase of the cell cycle (Litovchick et al, 2007a). *BRCA1*, *RAD51*, *RAD54*, and *BLM* helicase are all DREAM complex targets, which have been determined either experimentally or via in silico analysis (Sadasivam and DeCaprio, 2013; Muller et al, 2014; Fischer et al, 2016).

Here, we provide evidence that apart from the "canonical" p53-p21 axis, repression by the DREAM complex is also mediated by p53 via a "non-canonical" pathway. Hence, repression of the DREAM targets is observed in cells with wild type genotype and cells lacking p21 (i.e., p21−/− cells) but not in cells lacking p53 (i.e., p53−/− cells). Further, both p53 WT or its Transcriptionally Dead Mutant, p53 [(22,23,53,54)] mutant (henceforth called p53 TAD mutant), are both almost equally capable of recruiting the DREAM complex to its multiple target promoters. This process of

[1]Biotechnology Research and Innovation Council-National Institute of Immunology (BRIC-NII), Aruna Asaf Ali Marg, New Delhi 110067, India. [2]Biotechnology Research and Innovation Council-National Institute of Biomedical Genomics (BRIC-NIBMG), PO: NSS, Kalyani 741251, India. ✉E-mail: ssg2@nibmg.ac.in

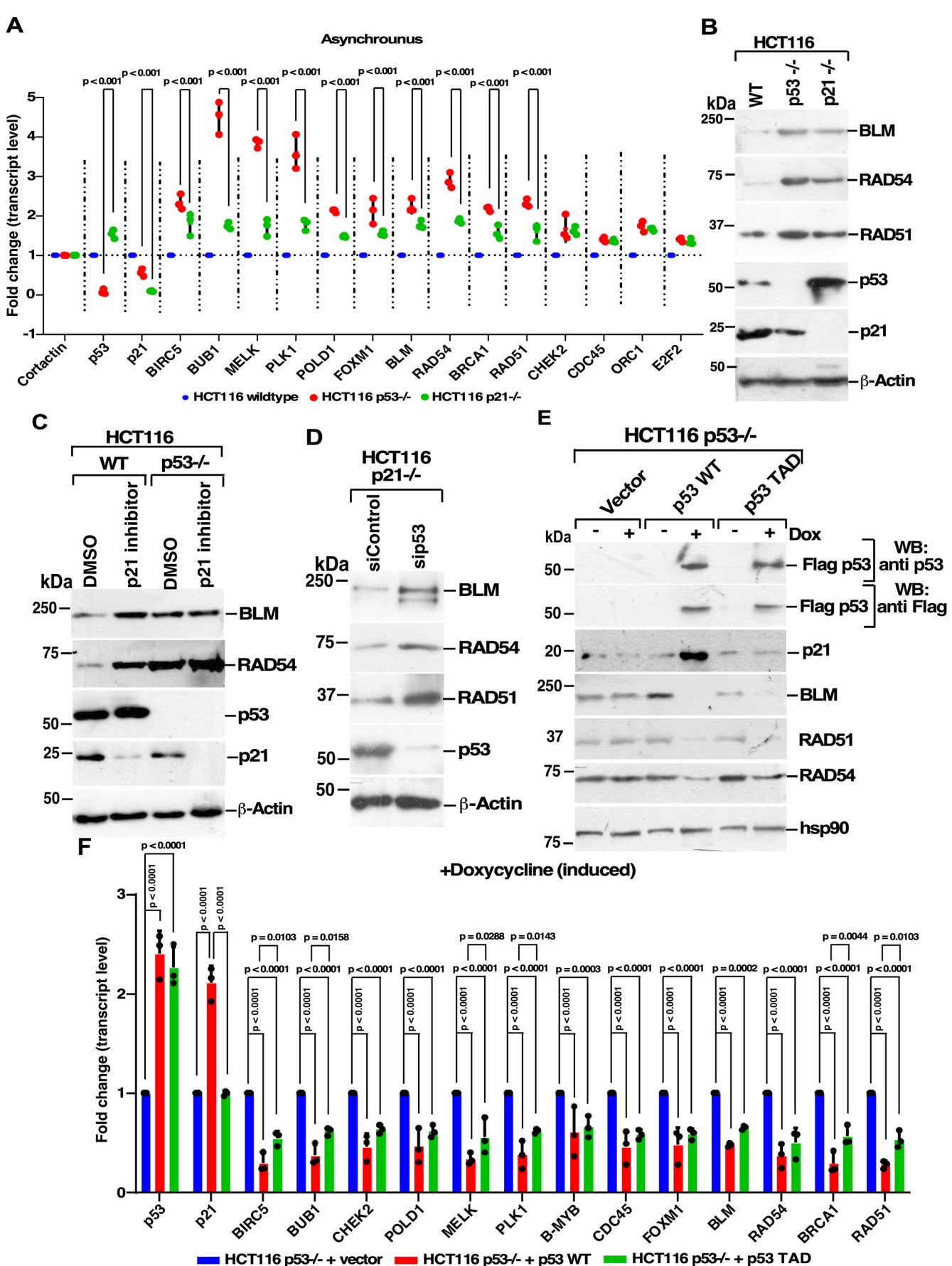

**Figure 1.   p53 can repress DREAM complex targets independently of p21.**

(**A**) Lack of p53 de-represses DREAM transcripts independently of p21. The transcript levels of the listed genes were determined by RT-qPCR from RNA isolated from HCT116 WT, HCT116 p53−/−, HCT116 p21−/− asynchronously growing cells. Cortactin was used as an internal control. Mean ± SD. The data is from three biological replicates. (**B**) Lack of p53 enhances the protein levels of DREAM complex targets as compared to lack of p21. Lysates were prepared from HCT116 WT, HCT116 p53−/−, HCT116 p21−/− cells. Immunoblotting was performed with the indicated antibodies. The data is from three biological replicates. (**C**) Repression of DREAM targets can also occur independently of the p53-p21 axis. Whole-cell lysates prepared from HCT116 WT, HCT116 p53−/− cells treated with either vehicle (DMSO) or p21 inhibitor. Immunoblotting was performed with the indicated antibodies. Three biological replicates were carried out and the same result was obtained. (**D**) p53 can repress DREAM targets in the absence of p21. HCT116 p21−/− cells transfected with either siRNA control or siRNA p53. Immunoblotting was performed with the indicated antibodies. Three biological replicates were carried out, and the same result was obtained. (**E, F**) Both p53 WT and p53 TAD mutant repress DREAM complex targets. (**E**) Lysates were prepared from the indicated stable lines grown in the absence or presence of Doxycycline (Dox). Western analysis was performed with the indicated antibodies. Mean ± SD. Three biological replicates were carried out, and the same result was obtained. (**F**) RNA was isolated from stable lines generated in HCT116 p53−/− cells expressing either the vector, p53 WT or p53 TAD mutant and grown in the presence of Doxycycline. RT-qPCR of the indicated genes was performed. The data is from three biological replicates. Source data are available online for this figure.

recruitment is dependent on the ability of p53 WT and p53 TAD mutant to interact with p130/p107, irrespective of their phosphorylation status. Using re-ChIP seq and the subsequent meta-analysis, we show that p53 is recruited along with the DREAM complex over the entire genome and regulates multiple pathways that lead to neoplastic transformation. Further, 2480 gene promoters in the presence of WT p53 and 2723 gene promoters either in the presence of p53 TAD mutant or in the absence of p21 have been identified as the new DREAM complex targets, which are regulated under asynchronous conditions via the "non-canonical" pathway. Overall, our study sheds new light on the regulation of the DREAM complex targets and provides an understanding of p53's role in transcriptional repression through its transactivation-independent function.

## Results

### p53 represses DREAM complex targets via two mechanisms

It is thought that the DREAM complex represses genes via the p53-p21 dependent axis (Fischer, 2017; Engeland, 2022). To elucidate whether the repression of the DREAM complex target can also occur in a p21-independent manner, we used three asynchronously growing isogenic human CRC cell lines, HCT116 p53+/+, HCT116 p53−/− and HCT116 p21−/− (Appendix Fig. S1A) which would mimic the physiological scenario in a vast majority of human tissues. In the absence of p53 and p21, the promoter activity of all four tested DREAM complex targets (BLM, RAD54, RAD51, and BRCA1) was enhanced. However, in all cases, the maximal promoter activity was observed in HCT116 p53−/− compared to HCT116 p21−/− (Appendix Fig. S1B,C). Loss of p53 and p21 both led to an increase in the transcript levels of all the tested known and tested (Muller et al, 2014; Bindra et al, 2005; Bindra and Glazer, 2007) or predicted (Fischer et al, 2016) DREAM complex targets (Fig. 1A). However, the lack of p53 alone always caused a greater de-repression of DREAM targets than the absence of only p21 in almost all the tested targets (Fig. 1A). These results indicated that while the "canonical" p53-p21-Rb axis can repress the DREAM complex function, p53 alone has an additional repressive effect on its target genes. Parallel western blot analysis in the same three isogenic lines also indicated increased protein levels due to the lack of p53 compared to p21 alone (Fig. 1B). Treatment of the same cell lines with a known p21 inhibitor, which leads to p21 degradation (Gupta et al, 2014), caused an increase in levels of BLM and RAD54 levels in only HCT116 WT cells but not in HCT116 p53−/− (Fig. 1C). Ablation of p53 in HCT116 p21−/− cells also led to further enhancement in the levels of three tested DREAM targets (Fig. 1D). Interestingly, the silencing of p27 but not p57 led to an increase in the expression of DREAM complex targets in HCT116 WT cells (Appendix Fig. S1D,E), probably due to the indirect role of p53 in regulating the p27-p21 axis (Tsoli et al, 2001). Overall, these results indicated the presence of a p21-independent, p53-dependent component mediating the repression of DREAM complex targets in asynchronously growing cells. We termed this effect as the "non-canonical" pathway of p53-mediated repression.

To determine whether the transactivation function of p53 was indeed essential to repress the DREAM complex targets, we utilized a known transactivation-dead p53 $^{(L22Q, W23S, W53Q, F54S)}$ mutant (p53 TAD mutant) (Lin et al, 1994). We created doxycycline-regulatable stable lines for both p53 WT and p53 TAD mutant in HCT116 p53−/− cells (Fig. 1E) which were grown under asynchronous conditions (Fig. S1F). The levels of p53 in doxycycline-regulatable stable lines expressing p53 WT and p53 TAD mutants are biologically relevant (1.25-fold increase) compared to the endogenous wild type p53 levels in HCT116 WT cells (Appendix Fig. S1G). Cell lines expressing either p53 WT or p53 TAD mutant could suppress the promoter activity of all tested DREAM complex targets in luciferase assays. However, the extent of repression was always less for the p53 TAD mutant compared to p53 WT (Appendix Fig. S1H, I). Further, expression of all the tested DREAM complex targets was repressed (at both transcript and protein levels) when either p53 wild type or p53 TAD mutant were expressed (Fig. 1E,F; Appendix Fig. S1J), thereby validating the presence of the transactivation-independent role of p53 in the repression of DREAM target genes.

Next, we wanted to explore whether the repression of the DREAM complex members occurs in a p53-dependent and p21-independent manner, even in response to DNA damage. We found that the three DNA-damaging drugs, namely Nutlin, Doxorubicin, and 5-Fluorouracil (5-FU), stabilized p53 and concurrently repressed BLM and RAD51 in both HCT116 WT and HCT116 p21−/− cells but not in HCT116 p53−/− cells (Appendix Fig. S2A–C). This indicated the existence of the "non-canonical" pathway of p53-mediated repression even in response to DNA damage.

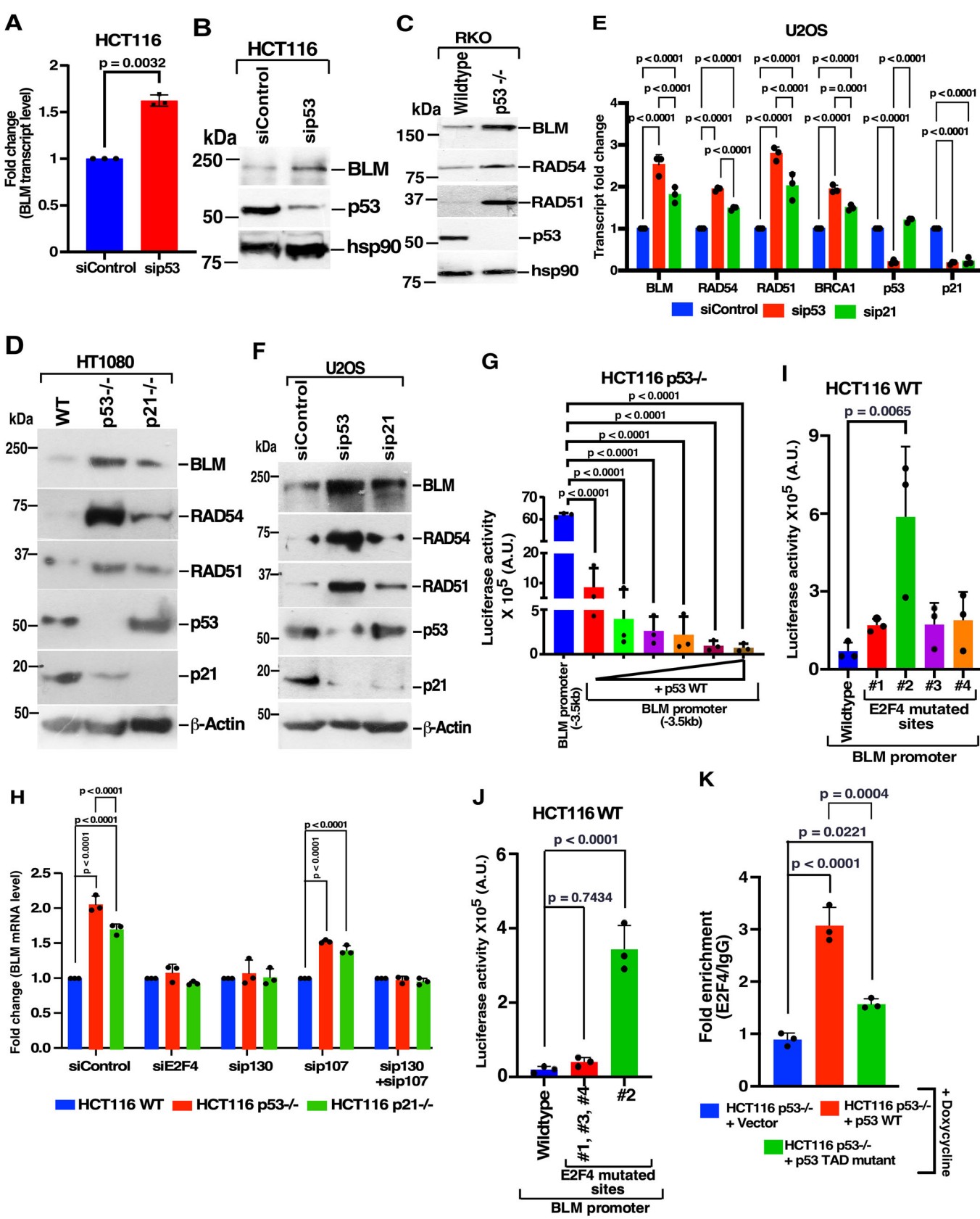

**Figure 2.   Transcription of BLM occurs via the p53-DREAM complex axis.**

(A–C) Transcript and protein levels of BLM were enhanced in the absence of p53. (A) HCT116 WT cells transfected with siRNA control or siRNA p53. RNA was isolated, and the transcript levels of BLM were determined by RT-qPCR. The data is from three biological replicates. Mean ± SD. (B, C) Lysates were prepared from either (B) HCT116 cells transfected with siRNA control or siRNA p53 or (C) RKO p53+/+ and RKO p53−/− cells. Immunoblotting was performed with the indicated antibodies. Three biological replicates were carried out, and the same result was obtained for both experiments. (D) Lack of p53 enhances the protein levels of DREAM complex targets more when compared to the lack of p21. Lysates were prepared from HT1080 WT, HT1080 p53−/−, and HT1080 p21−/− cells. Immunoblotting was performed with the indicated antibodies. Three biological replicates were carried out and the same result was obtained. (E, F) p53 can repress DREAM targets in the absence of p21. U2OS cells transfected with siRNA control, siRNA p53 or siRNA p21. (E) RNA was isolated, and RT-qPCR of the indicated genes was performed. The data is from three independent experiments. Cortactin was used as an internal control. Mean ± SD. (F) Immunoblotting was performed with the indicated antibodies. Three biological replicates were carried out, and the same result was obtained. (G) BLM promoter activity decreases in a p53 dose-dependent manner. Increasing amount of p53 (5 ng, 12.5 ng, 25 ng, 50 ng, 100 ng and 200 ng), pGL3-BLM promoter (−3.5 kb) and CMV-β-galactosidase were co-transfected into HCT116 p53−/− cells. Lysates made were used for luciferase and β-galactosidase assays. Mean ± SD. A.U. is absolute units after normalization with β-galactosidase activity. The data is from three biological replicates. (H) p53-dependent repression of BLM occurs via DREAM complex members. Transfection of the indicated siRNAs was carried out in HCT116 WT, HCT116 p53−/−, HCT116 p21−/− cells. RNAs were isolated and levels of BLM transcripts were determined by RT-qPCR. Mean ± SD. The data is from three biological replicates. (I, J) BLM repression is due to the #2 E2F site on its promoter. Basal BLM promoter activity was determined in HCT116 WT cells by luciferase assays after co-transfecting the cells with 640 bp pGL3-BLM constructs (wild type or the indicated mutant versions) and CMV-β-galactosidase. A.U. is absolute units after normalization with β-galactosidase activity. Mean ± SD. The data is from three biological replicates. (K) E2F4 binds to the #2 site on the BLM promoter in the presence of both p53 WT and p53 TAD mutant. Recruitment of E2F4 to the #2 site on BLM promoter was determined by ChIP-qPCR in stable lines generated in HCT116 p53−/− cells expressing either the vector, p53 wild type, p53 TAD mutant and grown in the presence of Doxycycline. Mean ± SD. The data is from three biological replicates. Source data are available online for this figure.

## p130/E2F4 play key roles in transcriptional repression by p53

Based on the above results, we wanted to understand the mechanism by which p53 repressed the DREAM complex promoters. Loss of p53 increased BLM at both transcript and protein levels in both HCT116 (Fig. 2A,B; Appendix Fig. S3A) and RKO (Fig. 2C) cells. Further, in both HT1080 (Fig. 2D) and U2OS (Fig. 2E,F) cells, the loss of p53 enhanced DREAM complex targets more significantly than p21 loss at both RNA and protein levels. Reciprocally, overexpression of a gradient of p53 WT in HCT116 p53−/− cell lines (Appendix Fig. S3B) suppressed BLM transcription in a dose-dependent manner (Fig. 2G). To further understand whether the regulation of BLM by p53 at the transcriptional level, the levels of newly synthesized BLM mRNAs were determined in HCT116 p53 +/+ and HCT116 p53−/− cells using a Click-iT™ chemistry-based kit. Nascent BLM mRNA synthesis levels were higher in p53−/− cells than in p53+/+ cells in all three tested time points (Appendix Fig. S3C). Ablation experiments using siRNAs against three core DREAM complex components (E2F4, p130, p107) in the isogenic cells indicated p53-dependent repression of BLM transcripts in HCT116 WT and p21−/− cells was critically dependent on two key DREAM complex members, E2F4 and p130, and only partially on p107 (Fig. 2H; Appendix Fig. S3D–H).

In silico analysis and published data (Muller et al, 2014; Yang et al, 2013) have indicated that there are four putative E2F consensus sites in tandem repeats on the BLM promoter. ChIP experiments indicated maximum E2F4 recruitment in the presence of p53 WT compared to the absence of either p53 or p21 (Appendix Fig. S3I). While p130 also recruits on BLM promoter at the #2 site in the presence of p53, the presence of p107 was negligible. It is noted that both E2F4 and p130 were still recruited on BLM promoter at the #2 site even in the absence of p21, though their recruitment was less as compared to the presence of p53 (Appendix Fig. S3I). To further decipher which E2F site(s) on BLM promoter was indispensable for BLM repression, 640 bp minimal promoter-luciferase constructs were constructed for wild type, single mutant and multiple-mutant versions of BLM promoter. BLM promoter

activity was maximally enhanced when the #2 binding site for E2F was mutated (Fig. 2I,J). ChIP assays done in the inducible stable lines with anti-E2F4 antibody confirmed the recruitment of E2F4 to the #2 E2F binding site on BLM promoter for both p53 WT and p53 TAD mutant (Fig. 2K). It is hypothesized that the #2 E2F4 binding site on the BLM promoter is probably preferred as it consists of an E2F consensus site (GCGGGAA) (Muller et al, 2012). p53 ChIP performed in HCT116 isogenic cells under asynchronous conditions, revealing basal-level recruitment of p53 to the #2 site on the BLM promoter. To further confirm p53 recruitment, cells were treated with DNA damaging agent 5-FU, which led to enhanced p53 recruitment at the #2 site of the BLM promoter in both HCT116 WT and HCT116 p21−/− cells (Appendix Fig. S3J).

## Both p53 WT and p53 TAD bind to p107/p130 on BLM promoter

To investigate how the two p53 variants cause repression of DREAM complex targets, we first conducted DNA affinity experiments using a biotinylated probe encompassing the 640 bp minimal BLM promoter and nuclear extracts from the three isogenic cell lines. Post-reaction, western blot analysis blot revealed that p53 and two key DREAM complex components (E2F4, p130) were enriched on the BLM promoter only in HCT116 p53+/+ and HCT116 p21−/− cells but not in HCT116 p53−/− cells. However, recruitment of p53, E2F4, p130 was less in HCT116 p21−/− compared to HCT116 p53+/+ cells (Fig. 3A). Similar DNA affinity experiments carried out using HT1080 isogenic cell lines showed the enrichment of p53 and the two DREAM complex components (p130 and E2F4) to the BLM promoter only in HT1080 WT and HT1080 p21−/− cells (Fig. 3B). The recruitment of E2F4, p130 and p53 to the BLM promoter was dependent on the #2 E2F binding site (Fig. 3C). It is to be noted that the recruitment of E2F4 on the mutant #2 site was partially reduced. It is possible because E2F sites are present on the BLM promoter in tandem repeats, and the recruitment of classical DREAM complex components is not solely dependent on DNA binding factors like E2F4. DNA affinity experiments using nuclear extracts of isogenic doxycycline-inducible stable cell lines also revealed that both p53 WT and p53 TAD mutant

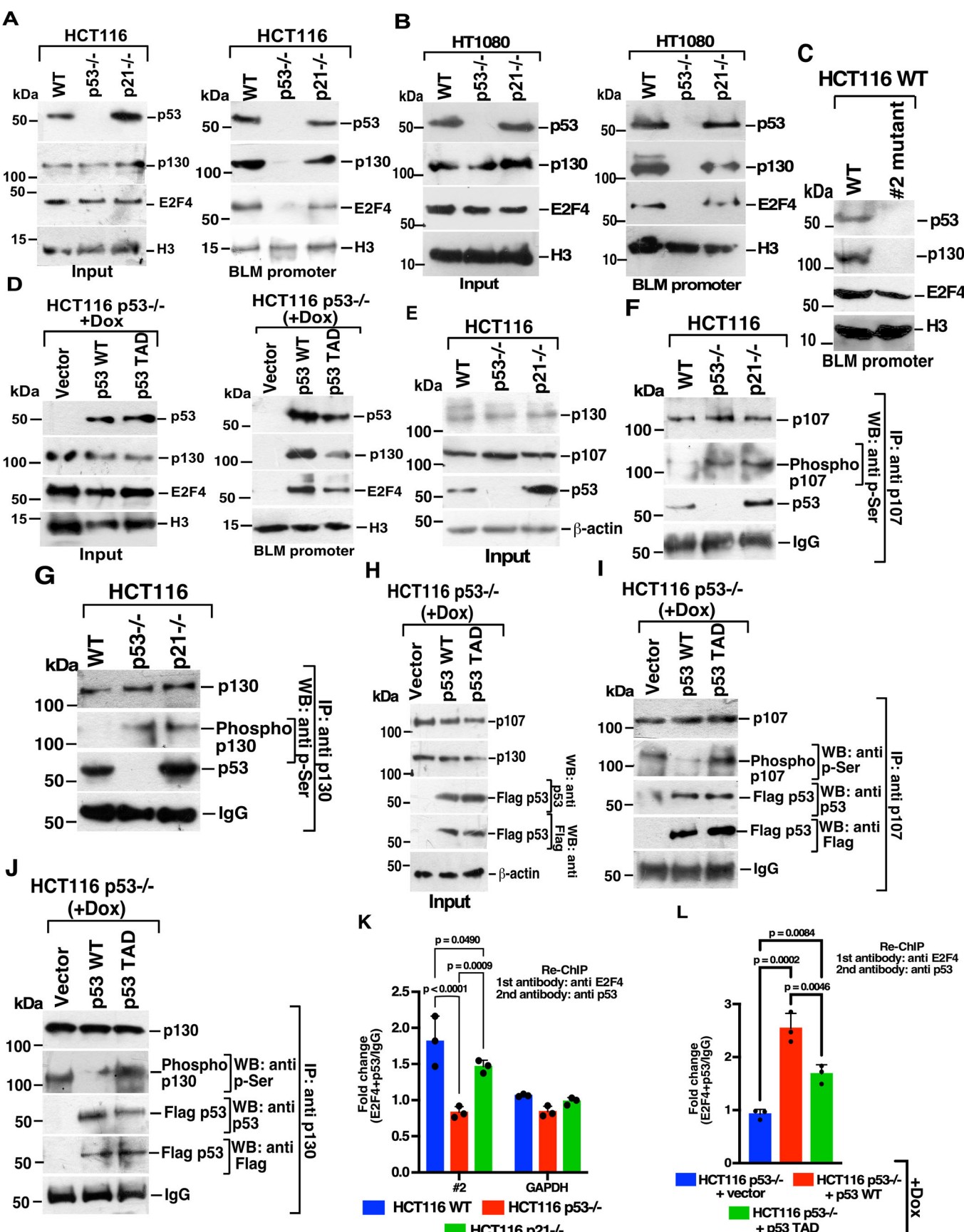

◄ **Figure 3. Both p53 WT and p53 TAD mutants bind to the BLM promoter.**

(A, B) p53 WT, along with DREAM complex members, bind to BLM promoter. (Left) Input for DNA affinity assay. Nuclear extracts were isolated from asynchronously growing (A) HCT116 WT, HCT116 p53−/−, HCT116 p21−/− cells, (B) HT1080 WT, HT1080 p53−/−, HT1080 p21−/− cells. Western blots were carried out with the indicated antibodies. (Right) DNA affinity assay was carried out using a biotinylated probe encompassing the 640 bp BLM minimal promoter as the "bait" and the nuclear extracts from (A) HCT116 p53 WT, HCT116 p53−/−, HCT116 p21−/− cells, (B) HT1080 WT, HT1080 p53−/−, HT1080 p21−/− cells as the "prey". Immunoblotting was performed using the indicated antibodies. Three biological replicates were carried out, and the same result was obtained. (C) Binding of p53 and DREAM complex members occur on the #2 E2F site in the BLM promoter. Same as (A), except the DNA affinity assay was carried out only in HCT116 p53 WT cells using a biotinylated probe encompassing either the 640 bp wild type BLM minimal promoter or BLM promoter where #2 E2F binding site is mutated. Three biological replicates were carried out and the same result was obtained. (D) Both p53 WT and p53 TAD mutant bind to BLM promoter. Same as (A) except the nuclear extracts were made from HCT116 p53−/− cells expressing p53 WT or p53 TAD mutant. Immunoblotting was performed using the indicated antibodies. Three biological replicates were carried out and the same result was obtained. (E–G). p107/p130 interact with p53 in cells even in the absence of p21. (E) Lysates were made from HCT116 WT, HCT116 p53−/−, HCT116 p21−/− cells. Immunoblotting was carried out with the indicated antibodies. (F, G) Immunoprecipitation was carried out with antibodies against either (F) p107 or (G) p130. Immunoprecipitates were probed with the indicated antibodies. Three biological replicates were carried out and the same result was obtained. (H–J) p107/p130 interact with both p53 WT and p53 TAD mutant irrespective of their phosphorylation status. (H) Lysates were made from inducible stable lines generated in HCT116 p53−/− cells expressing either the vector, Flag-tagged p53 WT or Flag-tagged p53 TAD mutant grown in the presence of Doxycycline. Immunoblotting was carried out with the indicated antibodies (I, J) Immunoprecipitation was carried out with (I) anti-p107 or (J) p130 antibodies, and immunoprecipitates were probed with the indicated antibodies. Three biological replicates were carried out, and the same result was obtained. (K) E2F4 and p53 are co-recruited onto the BLM promoter in the absence of p21. Re-ChIP assay was carried out using HCT116 p53 WT, HCT116 p53−/−, HCT116 p21−/− cells using anti-E2F4 as the 1st antibody and anti-p53 as the 2nd antibody. Co-recruitment of E2F4 and p53 on the #2 E2F site on the BLM promoter was determined by ChIP-qPCR analysis. E2F4-p53 co-recruitment to the GAPDH promoter was used as a control. Mean ± SD. The data is from three biological replicates. (L) E2F4 and p53 are co-recruited onto the BLM promoter in cells expressing either p53 WT or p53 TAD mutant. Similar to (K), recruitment of E2F4 and p53 to the #2 E2F site on BLM promoter was determined in the indicated stable lines grown in the presence of Doxycycline. Mean ± SD. The data is from three biological replicates. Source data are available online for this figure.

appreciably bound to BLM promoter along with p130 and E2F4 (Fig. 3D).

Since both p53 WT and p53 TAD mutant are recruited to the BLM promoter along with p130 and E2F4, we next aimed to determine whether the key DREAM complex subunits (E2F4, p130, p107) interacted with p53 WT and p53 TAD mutant. In vitro interaction assays were carried out between in vitro transcribed and translated p130, p107, and E2F4 with GST-tagged recombinant p53 proteins. Both p130 and p107 (but not E2F4) interacted equally well with p53 WT and p53 TAD mutant (Appendix Fig. S4A–C). Further interaction assays between in vitro transcribed and translated p130/p107 and GST-tagged recombinant p53 protein domains revealed that both these proteins probably interact with the p53 DNA-binding domain (75–320aa) (Appendix Fig. S4D-S4F). Reciprocally, p53 binds to the spacer regions of p130/p107 spanning (559–771aa) in the case of p130 and amino acids (586–780aa) in the case of p107, which is located between pocket A and pocket B domains (Appendix Fig. S4G–L). p130/p107 immunoprecipitation performed in the three isogenic cell lines also found p53 expressed in HCT116 p21−/− cells can interact with p130/p107 irrespective of their phosphorylation status (Fig. 3E–G). Similar immunoprecipitations performed using inducible stable cell lines also revealed both p53 WT and p53 TAD mutant to interact with p130 and p107 (Fig. 3H–J). Interestingly, the levels of phosphorylated p130/p107 were low in the presence of p53 WT and high in the presence of the p53 TAD mutant (Fig. 3H–J). This indicates that despite the differential phosphorylation status of p130/p107, both p53 WT and the p53 TAD mutant could interact with p130/p107, suggesting that the transcriptional activity of p53, i.e., upregulation of p21 and its function as Cyclin-dependent Kinase Inhibitor is dispensable for it (i.e., p53) to be recruited to the BLM promoters.

The above results provided the mechanistic basis of how p53 is capable of repressing the DREAM complex target promoters via a "non-canonical" pathway. Sequential ChIP (Re-ChIP) performed in HCT116 and HT1080 isogenic cells with antibodies against E2F4 followed by p53 indeed indicated that both E2F4 and p53 were

enriched on the BLM promoter in two isogenic lines HCT116 or HT1060 p53 WT or p21−/− and inducible lines expressing p53 TAD (Fig. 3K,L; Appendix Fig. S5A). Finally, to determine whether p53 recruitment depends on p130 or p107, sequential E2F4-p53 ChIP was performed in HCT116 WT upon ablation of either p130 or p107. In the absence of p130, both E2F4 and p53 were not enriched to the BLM promoter (Appendix Fig. S5B), thereby indicating the importance of p130 for the recruitment of E2F4 and p53 on the BLM promoter.

## p53 cancer mutants sequester E2F4

Having determined the role of both p53 WT and p53 TAD mutant in repressing DREAM complex targets, we next wanted to determine the roles of the two most common p53 cancer mutants (R175H and R248Q). Luciferase assays carried out in HCT116 p53−/− cells indicated that unlike p53 WT or p53 TAD mutant, p53 R175H and p53 R248Q did not repress the promoter activity of multiple DREAM complex targets (Appendix Fig. S6A–D). Further, we created a doxycycline-regulatable cell line expressing p53 R175H. Unlike the cells expressing p53 WT or p53 TAD mutant, cells expressing R175H could neither transactivate the p53 target (like p21) nor repress any of the DREAM complex targets (Fig. 4A,B; Appendix Fig. S6E).

In an effort to determine why p53 cancer mutants did not repress DREAM complex targets, in vitro interaction was carried out between in vitro transcribed and translated E2F4 with the GST-tagged recombinant p53 proteins (Appendix Fig. S6F). E2F4 was found to interact strongly with R175H and R248Q, in sharp contrast to p53 WT (Fig. 4C). Immunoprecipitation performed using the doxycycline-inducible stable cell lines confirmed the strong interaction of p53 R175H with E2F4 in cells (Fig. 4D). Next, to determine the consequences of this interaction, a DNA affinity experiment on the BLM promoter was carried out using nuclear extracts from isogenic lines expressing either p53 WT, p53 R175H or p53 R248Q. In contrast to p53 WT, the recruitment of p53 R175H or p53 R248Q was completely abolished, while that of p130

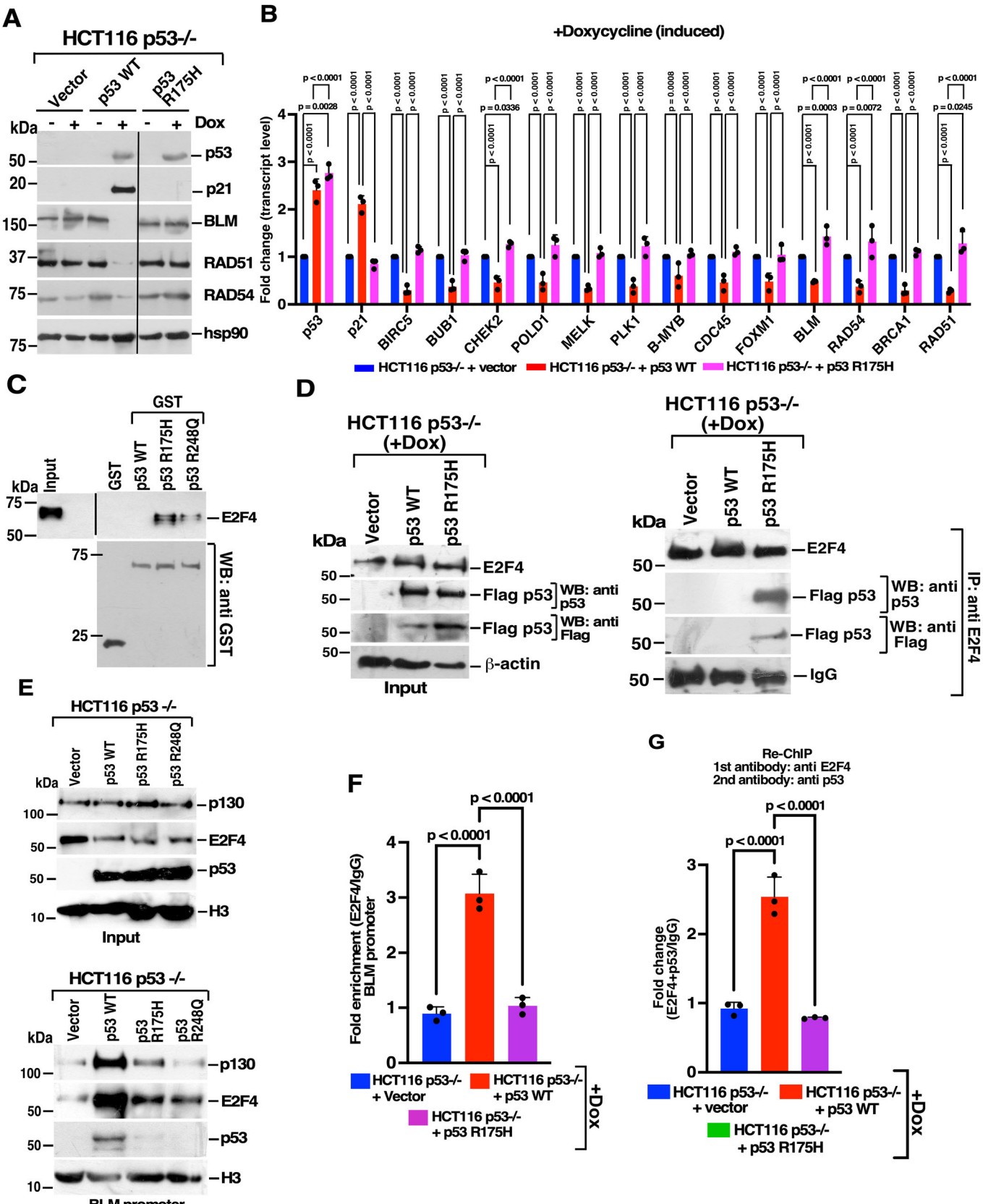

**Figure 4.   Cancer-associated p53 mutants do not repress DREAM complex targets.**

(A, B) p53 R175H cannot repress DREAM complex targets. (A) Lysates were prepared from the stable lines grown in the absence or presence of Doxycycline (Dox). Western analysis was performed with the indicated antibodies. Mean ± SD. Three replicates were carried out and the same result was obtained. (B) RNA was isolated from the indicated stable lines grown in the presence of Doxycycline. RT-qPCR of the indicated genes was performed. The data is from three biological replicates. (C) E2F4 interacts with cancer-specific hotspots p53 mutants in vitro. Interaction assays were carried out using in vitro-translated E2F4 protein and GST-tagged recombinant p53 variants. The interactions were determined by immunoblotting with anti-E2F4 antibody. Three biological replicates were carried out and the same result was obtained. (D) E2F4 interact with p53 R175H in cells. (Left) Input for immunoprecipitation of p130, p107 with p53 WT and p53 R175H. Lysates were prepared from the inducible stable cell lines expressing vector, Flag-tagged p53 WT or Flag-tagged p53 R175H in the presence of Dox. Western blots are carried out with the indicated antibodies. (Right) Immunoprecipitation was carried out with anri-E2F4 antibodies and immunoprecipitates were probed with the indicated antibodies. Three biological replicates were carried out and the same result was obtained. (E) p53 hotspot mutants cannot bind to BLM promoter. (Left) Inputs for DNA affinity assay. Nuclear extracts were isolated from HCT116 p53−/− cells expressing either p53 WT, p53 R175H, or p53 R248Q in the presence of Dox. Western blots are carried out with the indicated antibodies. (Right) DNA affinity experiment was performed using a biotinylated probe encompassing the 640 bp wild type BLM minimal promoter. Nuclear extracts were made. Immunoblotting was performed using the indicated antibodies. Three biological replicates were carried out, and the same result was obtained. (F) E2F4 does not bind to the #2 E2F site on the BLM promoter in the presence of p53 R175H. Recruitment of E2F4 to the #2 E2F site on BLM promoter was determined by ChIP-qPCR in stable lines expressing either vector, p53 WT or p53 R175H, grown in the presence of Doxycycline. The data is from three biological replicates. Mean ± SD. (G) E2F4 and p53 are not co-recruited onto the BLM promoter in cells expressing p53 R175H. Re-ChIP assay was carried out using stable lines expressing either the Vector, p53 WT or p53 R175H, grown in the presence of Doxycycline. Anti-E2F4 was used as the 1st antibody and anti-p53 as the 2nd antibody. Co-recruitment of E2F4 and p53 on the #2 E2F site on the BLM promoter was determined by ChIP-qPCR analysis. E2F4-p53 co-recruitment to the GAPDH promoter was used as a control. Mean ± SD. The data is from three biological replicates. Source data are available online for this figure.

and E2F4 was significantly reduced (Fig. 4E). E2F4 ChIP carried out in inducible stable cell lines also indicated that the expression of the p53 R175H mutant prevented the recruitment of E2F4 to the BLM promoter (Fig. 4F), thereby preventing repression of DREAM complex targets. Sequential ChIP (Re-ChIP) revealed that the enrichment of both E2F4 and p53 occur in cells expressing p53 WT but not in cells expressing p53 R175H, thereby confirming the inability of p53 cancer hotspot mutants to repress DREAM complex targets (Fig. 4G).

## Genome-wide recruitment of p53 on DREAM complex promoters

To determine whether p53 is indeed recruited at a genome-wide scale onto the DREAM complex target promoters, E2F4-p53 Re-ChIP-seq in HCT116 WT, HCT116 p21−/− and HCT116 p53−/− p53 TAD mutant cells grown in the presence of doxycycline was performed. Circos plots indicated genome-wide E2F4-p53 co-recruitment (Fig. 5A–C). The binding sites of E2F4-p53 co-recruitment in the presence of WT p53 in HCT116 WT were predominantly clustered near their respective Transcription start sites (TSS) (Fig. 5D). Next, multi-variant data analysis was carried out between E2F4-p53 Re-ChIP-seq in HCT116 p53 WT, Fischer DREAM targets (Fischer et al, 2016) and E2F4 ChIP (Zou et al, 2022). It was found that E2F4-p53 was co-recruited to 321 out of 1331 known Fischer DREAM complex targets. Interestingly, another 2480 targets were common between E2F4-p53 Re-ChIP and E2F4 ChIP, which were not Fischer DREAM complex targets. These 2480 targets are possibly regulated in a DREAM complex-dependent manner by the "non-canonical" pathway (Fig. 5E). Combined meta-analysis of the three cells expressing p53 (i.e., HCT116 p53 WT, HCT116 p21−/− and the inducible cell line expressing p53 TAD mutant) reveals the presence of 399 genes which are exclusively regulated by p53 via the "non-canonical pathway" and are independent of the p53-p21 transcriptional axis. More importantly, p53 in HCT116 p21−/− and p53 TAD mutant expressing cells were recruited to 2723 sites (Fig. 5F). Investigating the individual promoter regions, we found that the enrichment regions obtained in the E2F4-p53 Re-ChIP seq data extensively

overlapped with publicly available E2F4 ChIP seq data on BLM, RAD54, BUB1 and MELK promoters (Fig. 5G; Appendix Fig. S7A–C). In contrast, publicly available p53 ChIP-seq data did not detect p53 recruitment on these promoters, likely due to the very low levels of p53 recruitment under asynchronous conditions and the lower sensitivity of ChIP-seq compared to ChIP-qPCR (as observed in Appendix Fig. S3J). Pathway analysis of all the E2F4-p53 Re-ChIP enriched genes indicated that multiple pathways that have a direct or indirect influence on cancer initiation and progression are targeted by this complex (Fig. 5H). Altogether, the results indicate that genome-wide recruitment of the DREAM complex to its target promoters via the p21-independent function of p53 leads to transcriptional repression, which possibly helps to maintain genomic integrity and cellular homeostasis.

## Discussion

This study presents evidence that aside from the repression carried out by the "canonical" DREAM complex pathway, repression can also be carried out by wild type p53, p53 expressed in p21−/− cells and p53 TAD mutant. This mode of repression by p53 via the "non-canonical" pathway is achieved when p53 is recruited to some of the DREAM complex target promoters in a p21-independent manner (Fig. 6). We provide evidence that this mechanism occurs because p53 interacts with the two most critical DREAM complex components, p107/p130, irrespective of their phosphorylation status. The recruitment of p53 to the DREAM complex possibly enhances the repertoire of promoter selections and may also increase the stability of this complex. This "non-canonical" mechanism has been demonstrated not only for some DREAM complex targets (BLM, RAD51, RAD54, BRCA1) but also on a pan-genomic level through E2F4-p53 Re-ChIP-seq and subsequent meta-analysis of the data. Thus, the results highlight the transactivation-independent role of p53 in mediating transcriptional repression via the DREAM complex.

To identify the p21 independent role of p53 in repressing the DREAM complex, we utilized three complementary approaches— (a) usage of a specific p21 inhibitor (UC2288) which degrades p21,

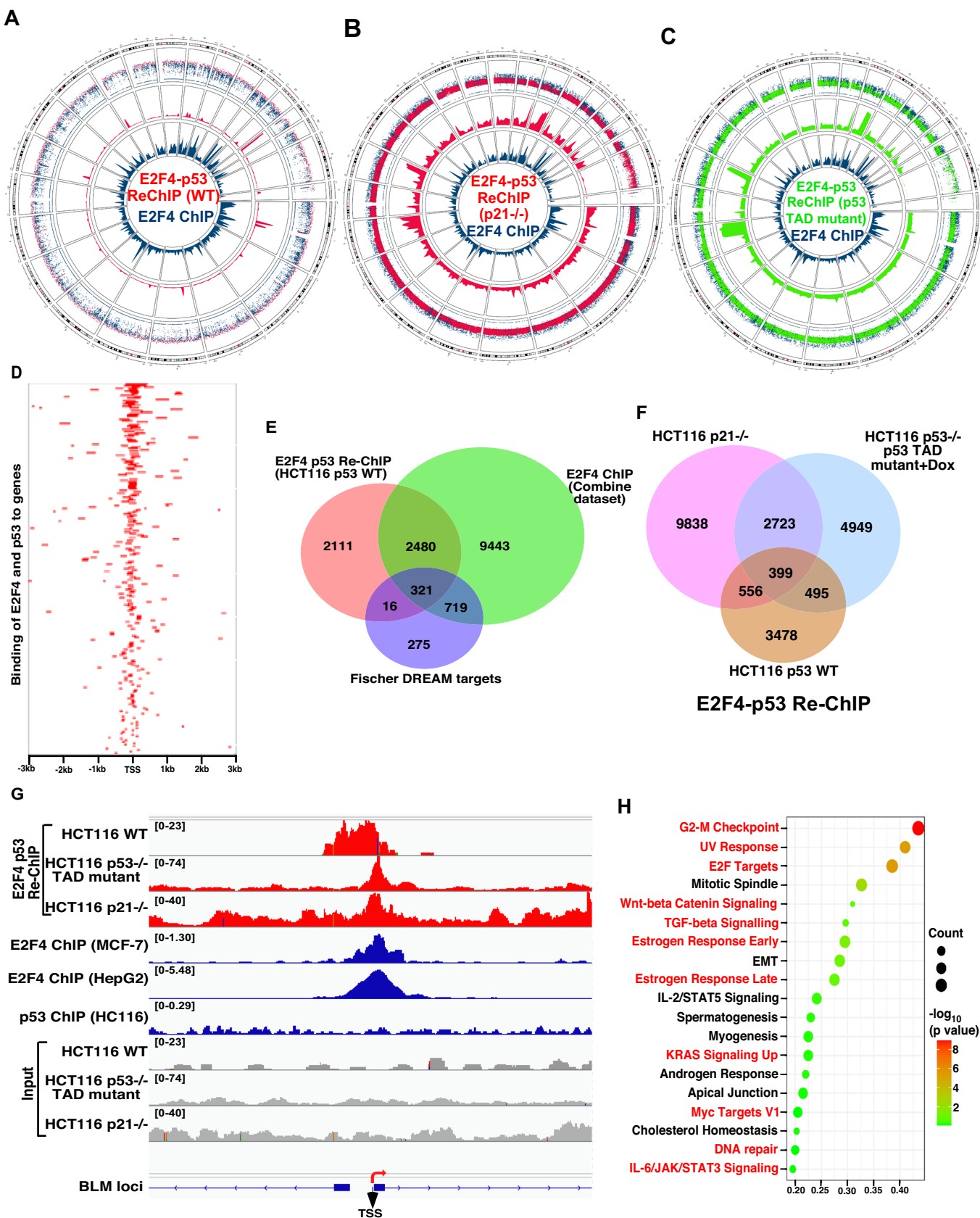

◄  **Figure 5.  Genome-wide co-recruitment of E2F4 and p53 on DREAM complex targets.**

(A–C) Circos plot representing the genome-wide co-recruitment profile of E2F4-p53. Circos plot was generated from E2F4-p53 ReChip-seq analysis carried out in (A) HCT116 p53 WT, (B) HCT116 p21−/−, (C) HCT116 p53−/− p53 TAD mutant (+Dox) cells and publicly available E2F4 ChIP seq datasets. The tracks from outside to inside sequentially represent: all the human chromosomes, E2F4-p53 Re-ChIP enrichment regions, E2F4 ChIP enrichment region, E2F4-p53 Re-ChIP intensity, and E2F4 ChIP intensity. (D) Heatmap representing E2F4-p53 recruitment in the presence of WT p53 near the TSS. Heatmap was generated based on E2F4-p53 Re-ChIP seq analysis carried out in HCT116 WT cells. (E, F) Multi-variant analysis of ChIP-seq. Analysis was done with (E) E2F4-p53 Re-ChIP-seq data from HCT116 p53 WT cells, Fischer DREAM targets, and E2F4 ChIP-seq data, (F) HCT116 p53 WT, HCT116 p21−/− and HCT116 p53−/− + p53 TAD mutant (+Dox) cells. Venn diagram was generated using peaks enriched within ±1 kb of the TSS in all cases. (G) E2F4, p53 Re-ChIP data overlap with E2F4 ChIP seq data. IGV browser tracks from E2F4 p53 Re-ChIP seq peaks (HCT116 p53 WT, HCT116 p21−/−, HCT116 p53−/− p53 TAD mutant (+Dox condition)), E2F4 ChIP seq peaks (HepG2, MCF7), p53 ChIP peak and respective input signals. Upstream of TSS for BLM promoter has been depicted. (H) Pathway analysis from E2F4-p53 Re-ChIP seq analysis indicates key pathways affected having implications during cancer progression. Pathway analysis of genes enriched in Re-ChIP-seq of HCT116 p53 WT was performed using GSEA MSigDB.

(b) HT1080, HCT116 isogenic cell lines (WT, p53−/− and p21−/− genotypes) and U2OS cells where p21 was depleted by siRNA (b) doxycycline regulatable cell lines in HCT116 p53−/− cells expressing either p53 WT or p53 TAD mutant. Both UC2288 and the isogenic system directly showed the effect of the lack of p21 on p53 function during the repression of DREAM complex targets. Importantly the p53 TAD mutant, which is known to be unable to induce cell-cycle arrest, apoptosis, colony suppression, or target gene expression in vitro (Zhu et al, 1998; Venot et al, 1999) or induce senescence or inhibit tumorigenesis in vivo (Brady et al, 2011; Jiang et al, 2011) also demonstrated that p53 alone can cause DREAM complex target repression.

BLM regulation primarily depends on p53, E2F4 and p130, with p107 playing a minor role. Interestingly, p53 WT interacts with both p130 and p107. This is likely due to the cell cycle-regulated expression of p130 and p107, where p130 is predominantly expressed in the G0/G1 phase and p107 in the late G1/S/G2 phase (Grana et al, 1998). In vivo, the interaction of p53 with p130 may occur in a cell cycle-specific manner, allowing p53 to be recruited to the target promoters.

The question that can be legitimately asked is why it has always been thought till now that the p53-p21 axis is the only way for the formation of the DREAM complex. This is probably because the present study has been done almost exclusively with asynchronously growing cells (which incidentally mimics the in vivo milieu in tissues), which has allowed us to discover a parallel cellular process by which p53 can repress the DREAM targets. Interestingly, cancer hotspot mutants of p53 sequester the DREAM complex by binding to its main DNA-binding subunit, E2F4, thereby preventing DREAM complex-mediated transcriptional repression. This function of the hotspot p53 mutants possibly contributes to its well-known characterized oncogenic role (Muller and Vousden, 2014) and may contribute to its gain-of-function (Alvarado-Ortiz et al, 2020) during carcinogenesis.

The meta-analysis after E2F4-p53 Re-ChIP seq of HCT116 p53 WT indicates a significant number of the Fischer DREAM targets (321 out of 1331, i.e., 24.1%) were validated. However, a substantially larger number of genes (exactly 4928) were obtained by the E2F4-p53 Re-ChIP seq analysis, out of which 2801 (i.e., 56.8%) had also been reported to be E2F4 targets in public databases. Out of these, 2480 genes are common between E2F4-p53 Re-ChIP seq and E2F4 ChIP and are not part of the Fischer DREAM target list. In future, these 2480 genes may be considered as DREAM complex targets via the "non-canonical" pathway, thereby hugely expanding the repertoire of DREAM complex-mediated gene regulation. Interestingly, Rb-E2F targets

like BRCA1 were also found to be regulated through the "non-canonical" DREAM pathway via p53-E2F4 co-recruitment (Bindra and Glazer, 2006).

Perhaps having greater significance is the meta-analysis of E2F4-p53 ReChIP-seq data from p53+/+, p21−/−, and p53 TAD mutant cells growing under asynchronous conditions. E2F4 and p53 were found to be commonly co-recruited to 399 targets across all three conditions. This finding suggests the existence of a genome-wide "non-canonical" pathway in which the existence of p53's transactivation function is redundant. Interestingly, comparing only p21−/− and p53 TAD mutant cells, E2F4 and p53 were found to be co-recruited to 2723 additional targets that are not targeted by p53 WT. This phenomenon possibly indicates that the recruitment of transactivation-deficient p53 may serve as a compensatory mechanism via which the cells were trying to prevent the initiation or propagation of genomic instability which results in vivo due to the loss of p21 or the transactivation function of p53. Altogether the meta-analysis of the ChIP-seq experiments highlights the dynamic nature of p53's role in maintaining genomic integrity, even under conditions of impaired "canonical pathway".

In recent times it has been accepted as a paradigm that p53-dependent repression is indirect (Sullivan et al, 2018a). Multiple lines of evidence like the usage of multiple-reporter assays (Verfaillie et al, 2016), a meta-analysis of genomic data involving p53 (Fischer, 2017), and the usage of Global run-on-sequencing (GRO-seq) and RNA profiling (Allen et al, 2014) have led to this conclusion. It is to be noted that simultaneous testing of hundreds of enhancer-reporter constructs (as done in multiple-reporter assays) does not produce endogenous chromatin context. While GRO-seq is undeniably powerful, it maps binding sites of transcriptionally active RNA polymerase II—which is not the way this study is proposing p53 to cause the repression of DREAM complex targets. We provide evidence that at a genome-wide scale, p53 recruits on the repressive DREAM complex and in such sites, there is no role of RNA polymerase II. We further show that the p53 TAD mutant (which is completely defective in the p53 transactivation function) can do the same function as wild-type p53—thereby providing validation to this new role of p53 as a transcriptional repressor. The "canonical" and non-canonical" functions of p53 possibly act in parallel in tissues (possibly depending on the cell cycle stage of each cell). Hence it is quite possible that within individual cells, the "non-canonical" pathway is equally active—as also seen by the 2480 new targets being discovered. Overall, the discovery of this pathway has led to the identification of a new mechanism by which the cells try to maintain genome stability in human cells.

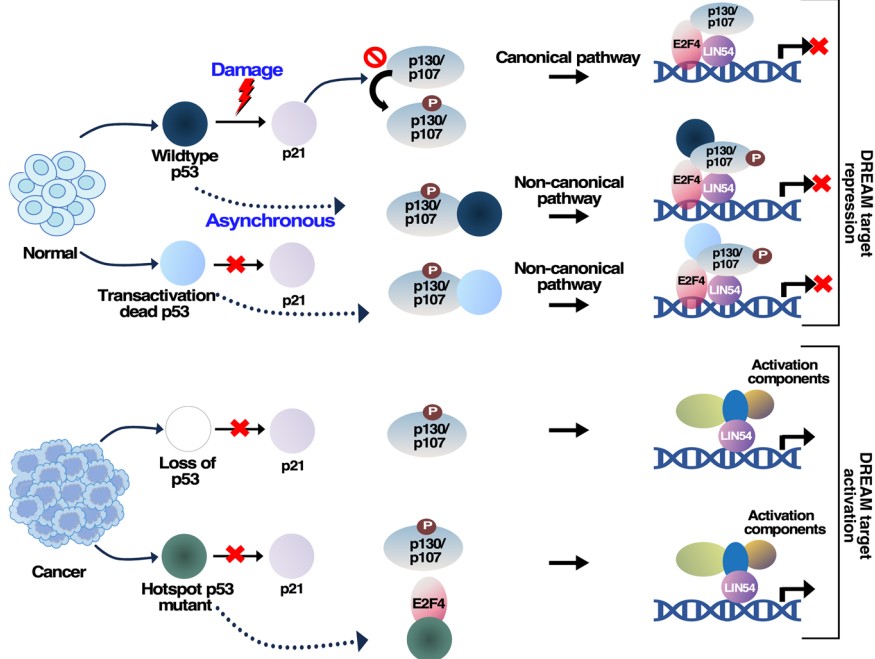

**Figure 6. p53 can repress DREAM complex targets in a p21-independent manner.**

p53 represses the DREAM complex by both "canonical" and "non-canonical" pathways. The expression of either wild type p53 or transactivation-dead p53 in normal cells presents two potential outcomes. In the first scenario, wild type p53 activates the cyclin-dependent kinase inhibitor p21, resulting in the hypo-phosphorylation of p107 and p130. This leads to the formation of the DREAM complex, which represses its targets through the "canonical pathway". However, both wild type and transactivation-dead p53 can interact with p130/p107 irrespective of their phosphorylation levels, through which p53 gets recruited on the DREAM complex promoters and represses the DREAM complex targets through the "non-canonical pathway". In cancer cells with bi-allelic loss of p53 or the presence of p53 "hotspot mutants"—different mechanisms come into play. Bi-allelic loss of p53 prevents the activation of p21, thus inhibiting the formation of the repressive DREAM complex. Similarly, p53 "hotspot mutants" physically bind and sequester E2F4, preventing the formation of the repressive DREAM complex. Hence in case of either bi-allelic loss of p53 or the presence of p53 "hotspot mutants,"—transcriptional activation occur of the DREAM complex targets.

# Methods

### Reagents and tools table

| Reagent/Resource | Reference or Source | Identifier or Catalog Number |
|---|---|---|
| IPTG | Merck | Cat# I6758; CAS Number 367-93- |
| DTT | Merck | Cat# D0632; CAS Number 3483-12-3 |
| PMSF | Merck | Cat# P7626; CAS Number 329-98-6 |
| Triton-X-100 | Merck | Cat# T9284; CAS Number 9002-93-1 |
| Ampicillin | Merck | Cat# A9393; CAS Number 69-53-4 |
| UC2288 (p21 inhibitor) | Merck | Cat# 5.32813 |
| Thiostrepton (FOXM1 inhibitor) | Merck | Cat# 598226; CAS Number 1393-48-2 |
| Doxycycline | Merck | Cat# D3447; CAS Number 10592-13-9 |
| Puromycin | Merck | Cat# P8833; CAS Number 58-58-2 |
| DMSO | Merck | Cat# D2650; CAS Number 67-68-5 |

| Reagent/Resource | Reference or Source | Identifier or Catalog Number |
|---|---|---|
| Lithium chloride | Merck | Cat# L4408; CAS Number 7447-41-8 |
| Sodium deoxycholate | Merck | Cat# D6750; CAS Number 302-95-4 |
| Proteinase K | Biobasic | Cat# PB0451; CAS Number 39450-01-6 |
| RNase | Merck | Cat# 10109134001 |
| Fetal bovine Serum | Thermo Fisher Scientific | Cat# 10082147 |
| Advanced DMEM | Thermo Fisher Scientific | Cat# 12491-023 |
| McCoy's 5A Medium | Thermo Fisher Scientific | Cat# 16600-108 |
| Optimem reduced serum medium | Thermo Fisher Scientific | Cat# 31985-088 |
| Qubit® dsDNA HS Assay Kit | Thermo Fisher Scientific | Cat# Q32851 |
| Lipofetamine 2000 | Thermo Fisher Scientific | Cat#11668019 |
| Lipofetamine 3000 | Thermo Fisher Scientific | Cat# L3000001 |
| LentiX concentrator | Takara | Cat# 631232 |

| Reagent/Resource | Reference or Source | Identifier or Catalog Number |
|---|---|---|
| Complete Protease Cocktail inhibitor | Roche | Cat# 11697498001 |
| BL21 (DE3) competent cells | Thermo Fisher Scientific | Cat# EC0114 |
| T7 Quick coupled Transcription/Translation system | Promega Corporation | Cat# L1170 |
| TRIzol Reagent | Thermo Fisher Scientific | Cat# 15596026 |
| Reverse transcriptase core kit | Eurogentec | Cat# RT-RTCK-03 |
| DyNamo color flash SYBR green qPCR kit | Thermo Fisher Scientific | Cat# F-416L |
| NE-PER™ Nuclear and Cytoplasmic Extraction Reagents | Thermo Fisher Scientific | Cat# 78833 |
| Pierce™ BCA Protein Assay Kits | Thermo Fisher Scientific | Cat# 23225 |
| Kapa Hyperprep Kit | Roche | Cat# 8105952001) |
| Phenol:Chloroform:Isoamyl Alcohol 25:24:1 Saturated with 10 mM Tris, pH 8.0, 1 mM EDTA | Merck | Cat# P3803 |
| Glutathione S-Sepharose | Cytiva | Cat# 17-5279-02 |
| **Experimental Models** | | |
| HEK293T | ATCC | Cat# CRL-3216 |
| HCT116 WT referred as HCT116 p53+/+ | Bert Vogelstein (Johns Hopkins Medicine, USA) | (Bunz et al, 1998) |
| HCT116 p53−/− | Bert Vogelstein (Johns Hopkins Medicine, USA) | (Bunz et al, 1998) |
| HCT116 p21−/− | Bert Vogelstein (Johns Hopkins Medicine, USA) | (Bunz et al, 1998) |
| HT1080 WT | Carol Prives (Columbia University, USA) | (Venkatesh et al, 2020) |
| HT1080 p53−/− | Carol Prives (Columbia University, USA) | (Venkatesh et al, 2020) |
| HT1080 p21−/− | Carol Prives (Columbia University, USA) | (Venkatesh et al, 2020) |
| RKO | Horizon | Cat# HD PAR-077 |
| RKO E6 referred as RKO p53−/− | Horizon | Cat# HD 106-002 |
| HCT116 p53−/− + pLVX-TetOne-Puro Vector | This study | N/A |
| HCT116 p53−/− + p53 WT | This study | N/A |
| HCT116 p53−/− + p53 TAD mutant | This study | N/A |
| HCT116 p53−/− + p53 R175H mutant | This study | N/A |
| U2OS | Present in the lab | ATCC Cat# HTB-96 |

| Reagent/Resource | Reference or Source | Identifier or Catalog Number |
|---|---|---|
| HEK293T | Present in the lab | ATCC Cat# CRL-3216 |
| **Recombinant DNA** | | |
| pGEX4T-1 p53 (1-393) referred as GST-p53 wild type | Present in the lab of corresponding author | (Sengupta and Wasylyk, 2001) |
| pGEX4T-1 p53 (1-75) referred as GST-p53 wild type | Present in the lab of corresponding author | (Sengupta and Wasylyk, 2001) |
| pGEX4T-1 p53 (76-320) referred as GST-p53 wild type | Present in the lab of corresponding author | (Sengupta and Wasylyk, 2001) |
| pGEX4T-1 p53 (321-362) referred as GST-p53 wild type | Present in the lab of corresponding author | (Sengupta and Wasylyk, 2001) |
| pGEX4T-1 p53 (363-393) referred as GST-p53 wild type | Present in the lab of corresponding author | (Sengupta and Wasylyk, 2001) |
| pGEX4T-1 p53 (L22Q, W23S, W53Q, F54S) referred as GST-p53 TAD mutant | This study | N/A |
| pcDNA3.1 hygro(+) p53 (1-393) referred as pcDNA3.1 hygro(+) p53 wild type | Ronald T. Hay (University of Dundee, UK) | (Rodriguez et al, 1999) |
| pcDNA3.1 hygro(+) E2F4 (1-413) (cloning sites HindIII and BamHI) | This study | N/A |
| pGL3-Basic BLM promoter −3499 to +64 with respect to TSS (cloning sites SacI and XhoI) referred as BLM promoter −3.5 kb | This study | N/A |
| pGL3-Basic BLM promoter −461 to +179 with respect to TSS (cloning sites SacI and XhoI) referred as BLM 640 bp minimal promoter | This study | N/A |
| pGL3-Basic RAD51 promoter −441 to +267 with respect to TSS (cloning sites NheI and HindIII) | This study | N/A |
| pGL3-Basic RAD54 promoter −557 to +154 with respect to TSS (cloning sites NheI and HindIII) | This study | N/A |
| pGl3-Basic BRCA1 promoter −497 to +274 bp with respect to TSS (cloning sites NheI and HindIII) | This study | N/A |
| pGL3 p21 luc | Wafik S. El-Deiry (Brown University, USA) | (el-Deiry et al, 1993) |
| pGL3 Cyclin G luc | Moshe Oren (Weizmann Institute of Science, Israel) | (Zauberman et al, 1995) |
| pGL3 Bax luc | John C. Reed (La Jolla Cancer Research Foundation, USA | (Miyashita and Reed, 1995) |

| Reagent/Resource | Reference or Source | Identifier or Catalog Number |
|---|---|---|
| pGL3 p53 con luc | Jerry Shay (University of Texas Southwestern Medical Center, USA) | (Funk et al, 1992) |
| pGL3 Mdm2 luc | Moshe Oren (Weizmann Institute of Science, Israel) | (Zauberman et al, 1993) |
| β galactosidase | Available in the lab of corresponding author | N/A |
| pLVX-TetOne-Puro-GFP | Jason Sheltzer (Cold Spring Harbor Laboratory, USA) | (Lukow et al, 2021) Addgene Cat#171123 |
| pLVX-TetOne-Puro-Flag p53 (1-393) (cloning sites EcoRI and BamHI) referred as pLVX-TetOne-Puro-Flag (+) p53 wild type | This study | N/A |
| pLVX-TetOne-Puro-Flag p53 (L22Q, W23S, W53Q, F54S) (cloning sites EcoRI and BamHI) referred as pLVX-TetOne-Puro-Flag (+) p53 TAD mutant | This study | N/A |
| pLVX-TetOne-Puro-Flag p53 R175H (cloning sites EcoRI and BamHI) referred as pLVX-TetOne-Puro-Flag (+) p53 R175H | This study | N/A |
| pcDNA3.1 + p130-HA (1-1139) | James A. DeCaprio (Dana-Farber Cancer Institute, USA) | (Litovchick et al, 2007b) |
| pcDNA3.1-HA-p107 (1-1068) | James A. DeCaprio (Dana-Farber Cancer Institute, USA) | (Litovchick et al, 2007b) |
| pcDNA3.1 + p130-HA (1-360) | This study | N/A |
| pcDNA3.1 + p130-HA (1-559) | This study | N/A |
| pcDNA3.1 + p130-HA (1-771) | This study | N/A |
| pcDNA3.1 + p130-HA (1-967) | This study | N/A |
| pcDNA3.1 + p107-HA (1-385) | This study | N/A |
| pcDNA3.1 + p107-HA (1-586) | This study | N/A |
| pcDNA3.1 + p107-HA (1-780) | This study | N/A |
| pcDNA3.1 + p107-HA (1-950) | This study | N/A |
| pcDNA3.1 hygro (+) BLM promoter E2F4 mutated site 1 (#1) | This study | N/A |
| pcDNA3.1 hygro (+) BLM promoter E2F4 mutated site 2 (#2) | This study | N/A |
| pcDNA3.1 hygro (+) BLM promoter E2F4 mutated site 3 (#3) | This study | N/A |

| Reagent/Resource | Reference or Source | Identifier or Catalog Number |
|---|---|---|
| pcDNA3.1 hygro (+) BLM promoter E2F4 mutated site 4 (#4) | This study | N/A |
| pcDNA3.1 hygro (+) BLM promoter E2F4 mutated site 1, 3 and 4 (#1 + #3 + #4) | This study | N/A |
| pMD2.G | Didier Trono (School of Life Sciences, Ecole Polytechnique Fédérale de Lausanne, Switzerland) | Addgene Cat#12259 |
| psPAX2 | Didier Trono (School of Life Sciences, Ecole Polytechnique Fédérale de Lausanne, Switzerland) | Addgene Cat#12260 |
| **Antibodies** | | |
| Rabbit anti-BLM (used for WB) | Bethyl Laboratories | Cat# A300-110A; RRID: AB_2064794 (WB: 1:2500) |
| Mouse anti-RAD54 (used for WB) | Abcam | Cat# ab10705 RRID: AB_297417 (WB: 1:1000) |
| Rabbit anti-RAD51 (used for WB) | Santa Cruz Biotechnology | Cat# sc-8349; RRID: AB_2253533 (WB: 1:2000) |
| Rabbit anti-hsp90 (used for WB) | Santa Cruz Biotechnology | Cat# sc-7947; RRID: AB_2121235 (WB: 1:2500) |
| Rabbit anti-p21 (used for WB) | Santa Cruz Biotechnology | Cat# sc-397; RRID: AB_632126 (WB: 1:1000) |
| Mouse anti-β-actin (used for WB) | Santa Cruz Biotechnology | Cat# sc-47778; RRID: AB_2714189 (WB: 1:2500) |
| Mouse anti-E2F4 (used for WB, ChIP, reChIP, reChIP seq) | Millipore | Cat# 05-312, RRID: AB_2097110 (WB: 1:2500, ChIP: 2 µg/reaction, reChIP: 3 µg/reaction, reChIP seq: 4 µg/reaction) |
| Mouse anti-p53 DO1 (used for WB in blots expressing p53 wild type and p53 R175H, ChIP, reChIP, reChIP seq) | Santa Cruz Biotechnology | Cat# sc-126, RRID: AB_628082 (WB: 1:5000, reChIP: 3 µg/reaction, reChIP seq: 4 µg/reaction) |
| Mouse anti-p53 PAb421 (used for WB in blots expressing p53 wild type and p53 TAD mutant, ChIP, reChIP, reChIPseq) | Santa Cruz Biotechnology | Cat# MABE283 (WB: 1:1000, reChIP: 3 µg/reaction, reChIP seq: 4 µg/reaction) |
| Rabbit anti-p107 (used for WB) | Proteintech | Cat# 13354-1-AP, RRID: AB_2238024) (WB: 1:1000) |
| Rabbit anti-p130 (used for WB) | Cell Signaling Technology | Cat# 13610, RRID: AB_2798274 (WB: 1:1000) |

| Reagent/Resource | Reference or Source | Identifier or Catalog Number |
|---|---|---|
| Rabbit anti-H3 (used for WB) | Abcam | Cat# ab18521, RRID: AB_732917 (WB: 1:5000) |
| Mouse Control IgG (used for ChIP, reChIP) | Santa Cruz Biotechnology | Cat# sc-2025, RRID: AB_737182 (ChIP: 2 µg/reaction, reChIP: 3 µg/reaction, reChIP seq: 4 µg/reaction) |
| Rabbit Control IgG (used for ChIP) | Abcam | Cat# ab46540; RRID: AB_2614925 (ChIP: 2 µg/reaction) |
| Mouse anti-BRCA1 (used for WB) | Abcam | Cat# ab16780; RRID: AB_2259338 (WB 1:2000) |
| Rabbit anti-p57 (used for WB) | Cell Signaling Technology | Cat# 2557, RRID: AB_2291591 (WB 1:1000) |
| Rabbit anti-p27 (used for WB) | Santa Cruz Biotechnology | Cat# sc-528, RRID: AB_632129 (WB 1:2000) |
| Rabbit anti-HA (used for WB) | Cell Signaling Technology | Cat# 5017, RRID: AB_10693385 (WB 1:4000) |
| Rabbit anti-GST (used for WB) | Thermo Fisher Scientific | Cat# 71-7500, RRID: AB_2533994 (WB 1:5000) |
| **Oligonucleotides and other sequence-based reagents** | | |
| siRNA sequence for p53 | Dharmacon | (Vikhanskaya et al, 2007) |
| siRNA sequence for E2F4 | Dharmacon | (Chen et al, 2002) |
| siRNA sequence for p130 | Dharmacon | Cat# J-003299-12-0020 |
| siRNA sequence for p107 | Dharmacon | Cat# J-003298-12-0050 |
| siRNA sequence for p21 | Dharmacon | (Hume et al, 2021) |
| siRNA sequence for p27 | Dharmacon | (Akashiba et al, 2006) |
| siRNA sequence for p57 | Dharmacon | Cat# J-003244-11-0020 |
| ON-TARGETplus Non-targeting siRNA #1 | Dharmacon | Cat# D-001810-01-05 |
| PCR primers | This study | Appendix Table S2 |
| **Chemicals, Enzymes and other reagents** | | |
| EcoRI HF | New England BioLabs | Cat# R0101L |
| HindIII HF | New England BioLabs | Cat# R0104L |
| BamHI HF | New England BioLabs | Cat# R0136L |
| SacI | New England BioLabs | Cat# R3156L |
| XhoI | New England BioLabs | Cat# R0146L |
| GST | This study | N/A |
| GST p53 (1-393) | This study | N/A |
| GST p53 (1-75) | This study | N/A |
| GST p53 (76-320) | This study | N/A |
| GST p53 (321-362) | This study | N/A |
| GST p53 (363-393) | This study | N/A |

| Reagent/Resource | Reference or Source | Identifier or Catalog Number |
|---|---|---|
| GST p53 TAD mutant | This study | N/A |
| GST p53 R175H | This study | N/A |
| GST p53 R248Q | This study | N/A |
| **Software** | | |
| GraphPad Prism 9 | GraphPad | https://www.graphpad.com/scientific-software/prism/ |
| IgV tool | IgV | https://igv.org/app/ |
| RStudio | RStudio Team | https://rstudio.com/ |
| Image J | Image J | https://imagej.net/Fiji |
| FlowJo | FlowJo | https://www.flowjo.com/solutions/flowj/ |

*WB* western blotting, *ChIP* chromatin immunoprecipitation, *reChIP* re-chromatin immunoprecipitation, *reChIP-seq* re-chromatin immunoprecipitation followed by sequencing, *N/A* not available.

## Cell lines and treatment

HCT116 p53+/+, HCT116 p53−/−, and HCT116 p21−/− cells were grown in McCoy's 5A media. RKO p53 + /+ and RKO E6 (referred as RKO p53−/−) cells were grown in RPMI-1640 media. U2OS, HEK293T, HT1080 WT, HT1080 p53−/−, HT1080 p21−/− cells were grown in DMEM media. In all cases the media were supplemented with 10% fetal bovine serum, L-glutamine or sodium pyruvate, and Penicillin, Streptomycin and Amphotericin B. Cells were maintained at 37 °C and 5% $CO_2$. Stable lines were generated in HCT116 p53−/− expressing doxycycline-inducible vector or wild type p53 or p53$^{(L22Q, W23S, W53Q, F54S)}$ [referred as p53 TAD] or p53 R175H. All stable lines were selected in 2 µg/ml puromycin and post-selection grown in 0.5 µg/ml puromycin. Induction with doxycycline (final concentration 0.5 µg/ml) was for 24 h. While the cells were treated with 10 µM p21 inhibitor (UC2288) for 6 h.

## Cell cycle analysis

Cells were harvested by trypsinization, washed with 1XPBS, and fixed with pre-chilled 70% ethanol for 30 min on ice. Cells were washed twice with cold 1XPBS, treated with RNase (10 µg/ml) for 15 min at room temperature and stained with propidium iodide (40 µg/ml). Cell cycle profiles were acquired using FACS Verse and analyzed by FlowJo software.

## Transfections

All transfections (both for siRNAs and plasmids) were carried out by Lipofectamine 2000 according to the manufacturer's protocol for 24–48 h. For western and transcript analysis 200 picomoles of siRNA and/or 50 ng–1 µg plasmids were transfected in 1:3 DNA:Lipofectamine ratio. For luciferase assays, 0.5 µg of the luciferase reporter and 0.5 µg β-galactosidase mammalian over-expression plasmids were co-transfected.

## Stable cell line generation

HEK293T cells were seeded in the 10-cm plate for virus production. At 70–80% confluency, co-transfection of lentiviral expression plasmids, namely pLVX-TetOne-Puro or pLVX-TetOne-Puro-p53 WT or pLVX-TetOne-Puro-p53 TAD or pLVX-TetOne-Puro-p53 R175H together with packaging plasmids (pMD2.G, pspax2) (in 1:1:1 plasmid molar ratio) were carried out using Lipofectamine 3000 (in 1:2 ratio). Media was changed 6 h post-transfection. Viral supernatants were collected 24 h and 48 h post-transfection and filtered using a 0.45 μm filter, followed by concentrating the virus overnight using a LentiX concentrator or PEG 8000 (in a 1:3 ratio), followed by centrifugation at $1600 \times g$ for 1 h. The viral pellet was dissolved in the TE buffer. Transduction was performed in HCT116 p53−/− cells in 12-well plates using polybrene (10 μg/ml). Media was changed 16 h post-transduction. Cells were selected with puromycin 2 μg/ml for 14days (Agrawal et al 2025).

## Purification of proteins

All GST-tagged proteins were expressed in *E. coli* BL-21 DE3 competent cells at 16 °C overnight and subsequently purified according to standard protocols by binding to Glutathione-S-Sepharose beads. The bound proteins were washed with 0.5 mM reduced glutathione. The bound proteins were used for the assays.

## In vitro interactions

Coupled in vitro transcription and translation were carried out for p130 or p107 using 1 mM cold methionine. Glutathione-S-Sepharose bead-bound target proteins were incubated with the in vitro translated interacting partner for 2 h at 4 °C. Post-reaction the mixture was centrifuged, and washed with GST buffer (50 mM Tris-Cl pH 7.5, 10 mM $MgCl_2$, 100 mM KCl, 5% glycerol-autoclaved, 0.5% NP-40). The products were electrophoresed using SDS-PAGE and subjected to immunoblotting.

## DNA affinity assay

For DNA affinity assay cells were grown in 15 cm plates to 80–90% confluency. Nuclei were isolated by the NE-PER™ Nuclear and Cytoplasmic Extraction Kit according to the manufacturer's guidelines. The protein concentration in the nuclear extract was determined by the BCA assay. Sonicated salmon sperm DNA was added to the nuclear extract to a final concentration of 100 μg/ml and incubated at room temperature for 15 min. Extracts were aliquoted into 1.5 ml Eppendorf tubes (with 3 mg protein/tube). 2 μg of biotinylated DNA probe (biotinylated DNA probes were amplified from the pGL3 vector containing the E2F4 binding sites on BLM promoter using biotin tagged primers) was added to the extracts and incubation continued for 30 min at room temperature. Subsequently, 80 μl of magnetic streptavidin bead suspension was added and the mixture was incubated for an additional 5 min. Proteins binding to the probes were isolated and washed three times with 500 μl wash buffer (20 mM HEPES-KOH, pH 7.9, 1.5 mM $MgCl_2$, 0.2 mM EDTA, 150 mM NaCl, 0.5% NP40, 1X PIC, 0.1 mM DTT). Proteins were eluted in an elution buffer (125 mM Tris pH 6.8, 10% SDS, 25% glycerol, 25% β-mercaptoethanol, 0.25%

bromophenol blue) by incubating at 95 °C for 10 min. 40 μl of the eluates and 20 μg nuclear extract (considered as input) were subjected to SDS PAGE and subjected to immunoblotting.

## Western blot

Cell lysates were prepared using the M2 lysis buffer (50 mM Tris HCl, pH 8.0, 100 mM KCl, 10 mM $MgCl_2$, 5% glycerol, 0.5% NP-40, 1X PIC). Tissue samples were crushed in liquid nitrogen, then tissue lysates were prepared in RIPA buffer (1 mM Tris HCl pH 7.8, 150 mM NaCl, 2% Triton X-100, 1%(w/v) Sodium deoxycholate, 0.1%(w/v) SDS supplemented with 1× PIC, 0.1 mM DTT, 1 mM sodium orthovanadate, 1 mM aprotinin, and 1 mM phenylmethylsulfonyl fluoride just before preparation of the lysates. In both cases, a reducing buffer (375 mM Tris HCl pH 6.8, 6% SDS, 48% glycerol, 9% β-mercaptoethanol and bromophenol blue) was added to the samples and boiled for 10 min. 40 μg of lysates were used for immunoblotting with the indicated antibodies.

## Immunoprecipitation

For each reaction, 500 μg of lysates with 2X PIC and 1 μg of the specific antibody (p130/p107/E2F4) was incubated overnight at 4 °C. 30 μl of Protein G beads slurry (equilibrated with 1X PBS + 0.1% NP-40) was added post-incubation for 3 h. Post-binding beads were washed thrice with 1 ml of 1X PBS + 0.1% NP-40. 6X SDS sample buffer was added to the beads and boiled for 12 min. Samples were resolved on 10% SDS-PAGE gel.

## Nascent RNA estimation by RT-qPCR

HCT116 p53+/+, HCT116 p53−/− cells were labeled with 0.5 mM 5-ethynyl uridine (EU) for 30, 45, and 60 min before collection. For the Click reaction, total RNA was collected and EU-labeled RNA was enriched by Dyna-beads following the manufacture protocol of Click-iT Nascent RNA Capture Kit. Total labeled pulled-down RNA was used for cDNA synthesis using the Reverse transcriptase core kit which was then used for RT-qPCR.

## ChIP and Re-ChIP analysis

For ChIP, $3 \times 10^6$ cells were cross-linked after trypsinization in situ with 1% formaldehyde for 10 min at 37 °C followed by quenching with 0.125 M glycine for 5 min at 37 °C. Post-wash with PBS, cells were resuspended in 1 ml of nuclear lysis buffer (1% SDS, 10 mM EDTA, 50 mM Tris–HCl pH-8.1), incubated on ice for 1 h and disrupted in a Diagenode bioruptor plus at 4 °C (30 s pulse, 30 s hold, 50 cycles). After verification that the sheared chromatin is of the right size (~200–400 bp), the samples were centrifuged at 13,000 rpm for 10 min at 4 °C and the supernatant was collected in a fresh tube. Chromatin (200 μg) was diluted 10 times with ChIP dilution buffer (1% Triton X-100, 2 mM EDTA, 20 mM Tris–HCl pH 8.1, 150 mM NaCl supplemented with 1X Protease Inhibitor Cocktail). Diluted chromatin was incubated with 3 μg of p130/p107/E2F4 antibody (or corresponding IgG) overnight at 4 °C with end-to-end shaking. The next day BSA (working concentration 1 μg/μl) and salmon-sperm blocked Protein A/G Sepharose beads were added and incubation continued for an additional 3 h, following which sequential washes (1 ml used each time) were

given with (a) Low salt wash buffer (0.1% SDS, 1% Triton X-100, 2 mM EDTA, 20 mM Tris HCl pH 8.1, 150 mM NaCl); (b) High salt wash buffer (0.1%SDS, 1% Triton X-100, 2 mM EDTA, 20 mM Tris HCl pH 8.1, 500 mM NaCl); (c) LiCl buffer (0.25 M LiCl, 1% sodium deoxycholate, 10 mM Tris–HCl pH 8.1, 1 mM EDTA, 2% IGEPAL); (d) TE buffer (10 mM Tris HCl pH 8.1, 1 mM EDTA) for 2 times. Finally, 150 µl of freshly prepared washing bead buffer (0.1 mM NaHCO$_3$, 1% SDS) was added to each sample along with proteinase K (1 µg/µl) and RNase A (5 µg/µl). Incubation was continued overnight at 65 °C to reverse the cross-link. DNA was recovered by Phenol: chloroform: isoamyl alcohol (25:24:1) extraction, followed by ethanol precipitation and one 70% ethanol wash. The final DNA was dissolved in 20 µl of 10 mM Tris HCl (pH 8.0) and used for ChIP-qPCR.

For Re-ChIP, after the final wash of the E2F4 ChIP with TE, the beads were resuspended in 75 µl TE with 2% SDS, 15 mM DTT and the immunocomplexes were eluted by incubating for 30 min at 37 °C. Centrifugation was done at 1200 rpm for 5 min, the supernatant was transferred and diluted 20 times with ChIP dilution buffer. 4 µg of p53 antibody was added to each sample and incubated overnight at 4 °C with constant rotation. Salmon-sperm blocked Protein A/G beads were added for the last 3 h, followed by sequential washes with 1 ml of Low salt buffer, High salt buffer, LiCl buffer and TE buffer. Further steps were proceeded as the E2F4 ChIP protocol to recover the DNA. Re-ChIP-qPCR and Re-ChIP seq was carried out with the eluted DNA (Furlan-Magaril et al, 2009).

### RT-qPCR, ChIP-qPCR, Re-ChIP qPCR

Total RNA was isolated from cells using the TRIZOL reagent containing 1% 2-mercaptoethanol. cDNA was generated using the Reverse transcriptase core kit according to the manufacturer's protocol. The concentrations of input samples, ChIP DNA and Re-ChIP DNA were determined by Qubit using a dsDNA HS assay kit. 1 ng of ChIP DNA or Re-ChIP DNA was used for qPCR with primers that amplify regions on BLM promoters. Recruitment onto the GAPDH promoter was used as an internal control. Real-time qPCR was carried out with the SYBR green qPCR kit using the QuantStudio 3 Real-time PCR system according to the manufacturer's instructions. The relative level of gene expression was determined using the $2^{(-\Delta\Delta Ct)}$ method.

### Re-ChIP sequencing and analysis

Re-ChIP DNA libraries were constructed using the Kapa Hyper-prep Kit with sample purification beads according to the manufacturer's instructions. Next-generation sequencing of libraries was performed using the NovaSeq 6000 S4 Reagent Kit v1.5 for the paired-end, 30 million reads. The analysis was carried out following the methodology outlined in a previous study (Kaur et al, 2024). Briefly, the quality of the raw reads was determined using FastQC (http://www.bioinformatics.babraham.ac.uk/projects/fastqc/), followed by removal of adapters using Trim Galore (https://www.bioinformatics.babraham.ac.uk/projects/trim_galore/). Bases with a high Phred score (more than 30) were aligned to the current human genome (hg38 assembly) using BWA (Li and Durbin, 2009). Peaks were detected using MACS2 and annotated using the ChIPseeker R package. Unique reads were then analyzed to determine the E2F4-p53 enrichment on TSS as described in

(Bardet et al, 2011). The data was visualized as a heatmap, Venn diagram, and Circos plot using RStudio and IGV tools. Pathway enrichment was performed using the Enrich R tool (Chen et al, 2013). Circos plot for E2F4 ChIP was created by merging all the breast cancer lines from the ChIP atlas. In the IgV plot, the E2F4 ChIP datasets used are MCF7_SRX3322323 and HepG2_SRX10478016 from the ChIP atlas. p53 ChIP dataset used: SRX2060918 from the ChIP atlas. E2F4 ChIP-seq targets were obtained from published literature (Fischer et al, 2016).

Dataset EV1 provides the following gene lists: 321 genes shared among E2F4 ChIP, E2F4-p53 Re-ChIP of HCT116 p53 WT, and Fischer DREAM targets (representing the "canonical" pathway); 2480 genes common to E2F4 ChIP and E2F4-p53 Re-ChIP (HCT116 p53 WT) but not in Fischer DREAM targets (representing the "non-canonical" pathway); 9443 genes exclusive to E2F4 ChIP, absent in both E2F4-p53 Re-ChIP (HCT116 p53 WT) and Fischer DREAM targets; and 2111 genes unique to E2F4-p53 Re-ChIP (HCT116 p53 WT), not found in E2F4 ChIP or Fischer DREAM targets.

Dataset EV2 contains the following gene lists: 399 genes shared across E2F4-p53 Re-ChIP of HCT116 p53 WT, HCT116 p21−/−, and HCT116 p53−/− p53 TAD mutant; 2723 genes common to E2F4-p53 Re-ChIP of p21−/− and p53 TAD mutant; 556 genes common to E2F4-p53 Re-ChIP of p53 WT and p21−/−; 495 genes common to E2F4-p53 Re-ChIP of p53 WT and p53 TAD mutant; 9838 genes exclusive to E2F4-p53 Re-ChIP of p21−/−; 4949 genes exclusive to E2F4-p53 Re-ChIP of p53 TAD mutant; and 3478 genes exclusive to E2F4-p53 Re-ChIP of p53 WT.

### Statistical analysis

All quantitations are presented as mean ± S.D. Experimental details are mentioned in the respective figure legends. The specific p values have been indicated in the figures. Details about the statistical test employed for each experiment are elaborated in Appendix Table S1 (1A, 1F; 2A, 2E, 2G–K; 3K, 3L; 4B, 4F, 4G; 5H, appendix figures). Quantification of Western blots for all three biological replicates is provided in Dataset EV3 as mean ± S.D.

## Data availability

The datasets produced in this study are available in the following database: E2F4-p53 Re-ChIP seq data of HCT116 p53 WT: Gene Expression in Array Express (accession no. E-MTAB-13359). The datasets produced in this study are available in the following database: E2F4-p53 Re-ChIP seq data of HCT116 p21−/− and HCT116 p53−/− p53 TAD mutant: Gene Expression in Array Express (accession no. E-MTAB-14629).

The source data of this paper are collected in the following database record: biostudies:S-SCDT-10_1038-S44318-025-00402-7.

## Peer review information

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

## Acknowledgements

The authors acknowledge Ronald T. Hay (University of Dundee, UK); Wafik S. El- Deiry (Brown University, USA); Moshe Oren (Weizmann Institute of Science, Israel); John C. Reed (La Jolla Cancer Research Foundation, USA); Jerry Shay (University of Texas Southwestern Medical Center, USA); James A. DeCaprio (Dana-Farber Cancer Institute, USA); Jason Sheltzer (Cold Spring Harbor Laboratory, USA); Didier Trono (School of Life Sciences, Ecole Polytechnique Fédérale de Lausanne, Switzerland) for recombinants; Bert Vogelstein (Johns Hopkins Medicine, USA); Carol Prives (Columbia University, USA) for cells. SS acknowledges funding from Biotechnology Research and Innovation Council-National Institute of Immunology (BRIC-NII) and Biotechnology Research and Innovation Council-National Institute of Biomedical Genomics (BRIC-NIBMG) core funds, Department of Biotechnology (DBT), India grant BT/PR41739/BRB/10/1974/2021, Board of Research in Nuclear Sciences (BRNS), India grant 58/14/17/2021-BRNS, Anusandhan National Research Foundation (ANRF) [Science & Engineering Research Board (SERB)], India grant CRG/2020/000125, Indo-French Centre for Promotion of Advanced Research (IFCPAR/CEFIPRA) grant IFC/6803-1/2022, J C Bose Fellowship JCR/2023/000020, Indo- French Centre for Promotion of Advanced Research (IFCPAR/CEFIPRA) grant IFC/6803- 1/2022 and Council of Scientific and Industrial Research (CSIR), India grant 27/0387/23/ EMR-II (SS).

## Author contributions

**Ritu Agrawal**: Conceptualization; Resources; Data curation; Formal analysis; Validation; Investigation; Methodology; Writing—original draft. **Sagar Sengupta**: Conceptualization; Resources; Supervision; Funding acquisition; Visualization; Writing—original draft; Writing—review and editing.

Source data underlying figure panels in this paper may have individual authorship assigned. Where available, figure panel/source data authorship is listed in the following database record: biostudies:S-SCDT-10_1038-S44318-025-00402-7.

## Disclosure and competing interests statement

The authors declare no competing interests.

