## [Peer Review File · The EMBO Journal]

p53 regulates DREAM complex-mediated repression in a p21-independent manner

Sagar Sengupta and Ritu Agrawal

Corresponding author(s): Sagar Sengupta (ssg2@nibmg.ac.in)

Review Timeline:

Submission Date:	23rd Jun 24
Editorial Decision:	16th Sep 24
Revision Received:	2nd Dec 24
Editorial Decision:	13th Jan 25
Revision Received:	18th Jan 25
Accepted:	10th Feb 25

Editor: *Cornelius Schneider*

Transaction Report:

Dear Dr. Sengupta,

Thank you again for submitting your manuscript to the EMBO Journal, for sharing your preliminary point-by-point reply and for our productive discussions. We think that the proposed revisions are reasonable and would therefore like to invite you to revise your manuscript based on your preliminary point-by-point reply.

I should also add that it is The EMBO Journal policy to allow only a single major round of revision and that it is therefore important to resolve the main concerns at this stage.

We generally allow three months as standard revision time, which can be extended to 6 months in case of major revisions, such as the experiments required here. As a matter of policy, competing manuscripts published during this period will not negatively impact on our assessment of the conceptual advance presented by your study. However, we request that you contact the editor as soon as possible upon publication of any related work, to discuss how to proceed. Should you foresee a problem in meeting the deadline, please let us know in advance and we may be able to grant an extension.

Thank you for the opportunity to consider your work for publication. I look forward to your revision.

Yours sincerely,

Cornelius Schneider

Cornelius Schneider, PhD
Editor
The EMBO Journal
c.schneider@embojournal.org

We realize that it is difficult to revise to a specific deadline. In the interest of protecting the conceptual advance provided by the work, we recommend a revision within 3 months (15th Dec 2024). Please discuss the revision progress ahead of this time with the editor if you require more time to complete the revisions. Use the link below to submit your revision:

Referee #1:

In this manuscript, Agarwal and Sengupta challenge the paradigm whereby the DREAM complex represses genes exclusively via the p53-p21 axis: upon stress, p53 activates transcription of the CDK inhibitor protein p21, which by inhibiting CDK, keeps p130/p107 in their hypophosphorylated state, thus facilitating DREAM complex formation. In this scenario, p53 plays an indirect role in DREAM-dependent transcriptional inhibition.

The authors reasoned that if p21 were solely responsible for p53-dependent DREAM complex formation and activity, loss of p21 should be sufficient to fully derepress DREAM complex-inhibited genes. However, by testing the effect of loss of either p53 or p21 on de-repression of known DREAM complex targets, they discovered that the strongest de-repression is only achieved upon p53 loss. Thus, the authors postulate that p53 can act in a DREAM-dependent/p21-independent manner to transcriptionally repress promoters of multiple DREAM-regulated genes, and the experiments presented in the manuscript are mostly supportive of this so called "non-canonical" p53-dependent pathway. In this pathway, p53 appears to be directly recruited to DREAM-inhibited promoters. Although well performed experimentally, this study leaves a number of questions to be addressed. To allow me to be able to recommend this work for publication in EMBO Journal, the authors would need to successfully address the issues below.

1. Data in Fig.1A-D show that for many DREAM-repressed genes, loss of p21 alone does not have as strong effect on their transcriptional derepression as does loss of p53. Furthermore, transcriptional repression of these genes is p53-dependent but does not require p53 to have functional trans-activation domain (TAD) (Fig.1E, F). Considering that p130/p107 components of the DREAM complex are supposed to be hypophosphorylated, is it possible that CDK inhibitors other than p21, such as p27 or p57, act in this pathway? This could be experimentally tested.
2. Systematic siRNA-mediated depletion of DREAM components (Fig.2E) shows that p53-dependent repression requires E2F4 and p130, while p107 appears to play a very minor role. Expression of p130 and p107 is cell cycle-regulated with p130 being mainly expressed in G0/G1 cells, and p107 in late G1/S/G2. Cell cycle profiles for HCT116-based cell lines used in the study (Fig.S1A) suggest that most asynchronously growing cells are in G1. This might explain why p130 appears to be more important compared to p107, i.e. it might be a cell cycle stage- and/or cell line-specific effect. The authors should the very least comment on this.
3. Expanding on the previous point, most of the mechanistic data in this manuscript are generated using the HCT116 cell line (except for Fig.2B, in which RKO WT and p53^{-/-} cells are used). It is important to demonstrate that the non-canonical p53/DREAM pathway is operating in a cell line-independent manner. Thus, I suggest that the authors carry out certain key assays in at least two other cell lines.
4. The detailed analysis of E2F4 and p130/p107 binding to BLM promoter presented in Fig.2F demonstrates neatly that DREAM complex is recruited to the strongest E2F consensus sequence within the BLM promoter. However, in this reviewer's opinion, it does not add much to the main message of the study. These data could therefore be moved to a supplementary figure.
5. The inputs for the IPs in Fig.3A and B, that are currently in Fig.S3A and S3B, should be moved to the main figure 3, as inputs are important controls for the IP experiments.
6. In the text on page 7, the authors state that "The recruitment of E2F4, p130 and p53 to the BLM promoter was dependent on #2 E2F binding site (Figure 3B)". Looking at the figure, p53 and p130 are indeed not recruited to the mutant BLM promoter, but E2F4 recruitment appears to be only slightly reduced. How do authors explain this discrepancy between the figure and the text? At the least, the text should be revised to address this issue.
7. GST pull down assays using in vitro transcribed/translated GST-p53, E2F4, p130 and p107 (Fig.S3C and D) show that p53 directly binds p130/p107 but not E2F4. This finding posits a couple of questions: a) which domain(s) of p53 and p130/p107 are important for the interaction? b) is p53 recruited to the BLM promoter upon depletion of p130 and p107. The manuscript will be enhanced if the authors could provide some data along these lines.
8. One of the intriguing findings of this study is that when either of the two most common p53 mutants is overexpressed, they are now able to interact with E2F4 (unlike the WT p53). These mutants are not recruited to the BLM promoter and, moreover,

their expression results in the major reduction in DREAM binding to the BLM promoter and, consequently, in defective DREAM-mediated transcriptional repression of relevant promoters. Both p53 mutants have previously been shown to spuriously bind to and cause aggregation of other proteins, e.g. p63 and p73. It is therefore possible that these p53 mutants exert similar activity towards E2F4. Based on all the data presented and described in Fig.4 and S4, the title of the section "E2F4 sequesters p53 cancer mutants" should probably be altered to something like "p53 cancer mutants sequester E2F4".

The interpretation that the "non-canonical" function of p53 is to recruit DREAM complex to the target promoters is not rigorously established. If those promoters do not contain p53 consensus binding sites then p53 is recruited there indirectly, i.e. via its interaction with DREAM complex. This is indeed what has been observed in this study. The Re-ChIP in Fig.5 and S5, is also done in a way of performing E2F4 ChIP first followed by p53 ChIP. Thus, a model in which the DREAM complex recruits p53 to (some) of its promoters is probably more accurate. The role of p53 might be in stabilising the DREAM complex at promoters rather than in recruiting it.

Referee #2:

One key mechanism by which the tumor suppressor p53 represses transcription has been proposed to involve the DREAM complex. Upregulation of the cyclin-dependent kinase inhibitor p21 by p53, in turn, causes dephosphorylation of key DREAM components, leading to suppressed gene expression. In this intriguing study, the authors present data to argue that p53 can repress via the DREAM complex in a manner independent of p21. And that this repression occurs with a transactivation-deficient mutant of p53. They propose a "non-canonical" pathway for DREAM-mediated repression that involves a direct interaction of the complex with p53.

Understanding mechanisms by which p53 regulates gene expression is a critically important area given the relevance of p53 as a tumor suppressor in human cancer. The notion that p53-dependent repression occurs in a p21-independent manner via the DREAM complex is a new and exciting finding. Thus, the manuscript has both high significance and novelty making it suitable for the readership of The EMBO Journal. However, the data as presented is too preliminary at this time and lacks essential controls experiments. The manuscript needs substantial revision before it is ready for publication. Key issues are as follows:

First, a key finding in the study is the p21-independence of the effects. The authors show data using the HCT116 cell line that has been engineered by homologous recombination to no longer express 21. These findings need to be validated in other cell lines using RNAi or other means to knockdown p21 expression to confirm that effects are not peculiar to a specific cell line.

Second, the use of the doxycycline-inducible expression of the transactivation-deficient p53 mutant is central to one of the key conclusions. It is unclear whether the levels of p53 that is achieved in this engineered system have biological relevance. Levels should be compared to cells with endogenous p53 expression to ensure that the expression is not artifactually high. This would be one explanation for the surprising finding that the transactivation-dead mutant retains activity in these assays.

Third, the ChIP-seq experiments need to be performed in the p21-null cells. An important finding is that the DREAM repression is p21-independent. The authors propose a mechanism to explain this involving p53 recruitment to genes by DREAM in the absence of p21. This needs to be directly demonstrated experimentally.

Fourth, many of the conclusions are supported by immunoblotting data. Since it is to be presumed that these blots were replicated, quantitation of multiple blots accompanied by some statistical analysis would strengthen conclusions, rather than merely showing a single replicate blot.

Fifth, much of the published literature has addressed a role for DREAM in the response of p53 to DNA damage. It is unclear how to integrate the findings shown here with what has been previously shown. This is important as the authors are proposing a "non-canonical" pathway. One possibility is that the p21-dependence of DREAM effects may be different basally versus in the DNA damage response. This needs to be addressed, so that the current findings can be clearly placed in the context of what has previously been published.

Sixth, given the model being proposed, it is confusing why occupancy by p53 is not revealed in p53 ChIPseq experiments. Although the model suggests that p53 is recruited by DREAM, one would expect p53 occupancy still to be detected by a direct ChIP analysis using a suitable p53 antibody.

Seventh the authors focus on four targets, BLM, RAD54, RAD51, and BRCA1 to show p21-independence. They need to show other targets that are p21-dependent and address what may be possible determinants for which pathway is utilized for repression. Is this target gene specific? And if so, is there a sense of why? It is unclear whether the findings in Figure 5 reflect the sensitivity of assays or that there are in fact distinct targets for p53 association.

Eighth, additional mechanistic insight is needed to explain the consequence of p53 in the complex that is assembled on specific promoters. The presence of the transactivation-dead mutant in this complex and how this generates a repression signal also

needs to be addressed. Why does the presence of p53 at these promoters in the context of DREAM result in repression? What is happening to the potent activation domain of p53 in this complex? The model presented in Figure 6 needs clarification. It is unclear what is the difference between the canonical and non-canonical pathways. What does the presence of p53 at the promoter add?

Additional comment:

1. In the Introduction, the authors state unequivocally that previous studies show that p53 represses indirectly via DREAM. There is a substantial published literature that supports the idea that, at least on some targets, p53 likely acts directly to repress.

Answers to reviewers' comments

Reviewer #1

In this manuscript, Agrawal and Sengupta challenge the paradigm whereby the DREAM complex represses genes exclusively via the p53-p21 axis: upon stress, p53 activates transcription of the CDK inhibitor protein p21, which by inhibiting CDK, keeps p130/p107 in their hypophosphorylated state, thus facilitating DREAM complex formation. In this scenario, p53 plays an indirect role in DREAM-dependent transcriptional inhibition.

The authors reasoned that if p21 were solely responsible for p53-dependent DREAM complex formation and activity, loss of p21 should be sufficient to fully derepress DREAM complex-inhibited genes. However, by testing the effect of loss of either p53 or p21 on derepression of known DREAM complex targets, they discovered that the strongest derepression is only achieved upon p53 loss. Thus, the authors postulate that p53 can act in a DREAM-dependent/p21-independent manner to transcriptionally repress promoters of multiple DREAM-regulated genes, and the experiments presented in the manuscript are mostly supportive of this so called "non-canonical" p53-dependent pathway. In this pathway, p53 appears to be directly recruited to DREAM-inhibited promoters. Although well performed experimentally, this study leaves a number of questions to be addressed. To allow me to be able to recommend this work for publication in EMBO Journal, the authors would need to successfully address the issues below.

Response: We thank the reviewer for the positive comments. In the revised version, we have tried to answer almost all the reviewers' queries.

1. Data in Fig.1A-D show that for many DREAM-repressed genes, loss of p21 alone does not have as strong effect on their transcriptional derepression as does loss of p53. Furthermore, transcriptional repression of these genes is p53-dependent but does not require p53 to have functional trans-activation domain (TAD) (Fig.1E, F). Considering that p130/p107 components of the DREAM complex are supposed to be hypophosphorylated, is it possible that CDK inhibitors other than p21, such as p27 or p57, act in this pathway? This could be experimentally tested.

Response: We agree with the reviewer's comment.

We performed p27/p57 siRNA mediated ablation experiment in HCT116 WT, HCT116 p53^{-/-} and HCT116 p21^{-/-} cells. the silencing of p27 but not p57 led to an increase in the expression of DREAM complex targets in HCT116 WT cells (Figure S1D, S1E), probably due to the indirect role of p53 in regulating p27-p21 axis (Tsolli *et al*, 2001).

2. Systematic siRNA-mediated depletion of DREAM components (Fig.2E) shows that p53-dependent repression requires E2F4 and p130, while p107 appears to play a very minor role. Expression of p130 and p107 is cell cycle-regulated with p130 being mainly expressed in G0/G1 cells and p107 in late G1/S/G2. Cell cycle profiles for HCT116-based cell lines used in the study (Fig.S1A) suggest that most asynchronously growing cells are in G1. This might explain why p130 appears to be more important compared to p107, i.e. it might be a cell cycle stage- and/or cell line-specific effect. The authors should the very least comment on this.

Response: We agree with the reviewer's comment.

In the discussion section, we stated that “BLM regulation primarily depends on p53, E2F4 and p130, with p107 playing a minor role. Interestingly, p53 WT interacts with both p130 and p107. This is likely due to the cell cycle-regulated expression of p130 and p107, where p130 is predominantly expressed in the G0/G1 phase and p107 in the late G1/S/G2 phase (Grana *et al*, 1998). In vivo, the interaction of p53 with p130 may occur in a cell cycle-specific manner, allowing p53 to be recruited to the target promoters” (Page 12).

3. Expanding on the previous point, most of the mechanistic data in this manuscript are generated using the HCT116 cell line (except for Fig.2B, in which RKO WT and p53^{-/-} cells are used). It is important to demonstrate that the non-canonical p53/DREAM pathway is operating in a cell line-independent manner. Thus, I suggest that the authors carry out certain key assays in at least two other cell lines.

Response: We appreciate the reviewer's comment.

We have performed the key experiments in the HT1080 isogenic cells (WT, p53^{-/-} and p21^{-/-} genotypes) and U2OS, where p53 and p21 have been ablated.

A. In U2OS cells, where p53 and p21 were ablated, RT-qPCR and western blot analyses revealed that similar to HCT116 cells, the levels of DREAM complex targets at both transcript and protein increased more significantly in the absence of p53 compared to the absence of p21 (Figure 2E, 2F).

B. In HT1080 isogenic cells, protein levels of DREAM complex targets are determined through western (Figure 2D), DNA affinity experiment on BLM promoter (Figure 3D) and E2F4-p53 Re-ChIP carried out and similar results were obtained as HCT116 cells (Figure S5A).

4. The detailed analysis of E2F4 and p130/p107 binding to the BLM promoter presented in Fig.2F demonstrates neatly that the DREAM complex is recruited to the strongest E2F consensus sequence within the BLM promoter. However, in this reviewer's opinion, it does not add much to the main message of the study. These data could therefore be moved to a supplementary figure.

Response: We agree with the reviewer's comment. We have moved Figure 2F to a supplementary Figure S3I.

5. The inputs for the IPs in Fig.3A and B, that are currently in Fig.S3A and S3B, should be moved to the main figure 3, as inputs are important controls for the IP experiments.

Response: We agree with the reviewer's comment. We have moved all the input of the IPs to the main figures 3A, 3D, 3E, 3H, 4D and 4E.

6. In the text on page 7, the authors state that "The recruitment of E2F4, p130 and p53 to the BLM promoter was dependent on #2 E2F binding site (Figure 3B)". Looking at the figure, p53 and p130 are indeed not recruited to the mutant BLM promoter, but E2F4 recruitment appears to be only slightly reduced. How do authors explain this discrepancy between the figure and the text? At the least, the text should be revised to address this issue.

We agree with the reviewer's comment. In Figure 3B, E2F4 recruitment appears to be partially reduced. It is because E2F sites are there in tandem repeats on the BLM promoter. We have changed the text accordingly in the result section of the manuscript (Page 8).

7. GST pull down assays using in vitro transcribed/translated GST-p53, E2F4, p130 and p107 (Fig.S3C and D) show that p53 directly binds p130/p107 but not E2F4. This finding posits a couple of questions: a) which domain(s) of p53 and p130/p107 are important for the interaction? b) is p53 recruited to the BLM promoter upon depletion of p130 and p107. The manuscript will be enhanced if the authors could provide some data along these lines.

Response: We appreciate the reviewer's comments and queries.

A. We have performed an in-vitro interaction assay of p130/p107 with different domains of p53. We found that p130 and p107 interact probably with the p53 DNA-binding domain (75-320aa) (Figure S4D-S4F).

B. We will also perform an in-vitro interaction assay of p53 with different domains of p130/p107. We observed that the p53 protein binds to the spacer regions of p130 (559-771aa) and p107 (586-780aa), spanning between pocket A and pocket B domains (Figure S4G-S4L).

C. We have performed an E2F4-p53 ReChIP on the BLM promoter after the shutdown of p130 and p107. Our findings indicate that E2F4 and p53 were not co-recruited to the BLM promoter upon p130 depletion but recruited upon p107 depletion. This suggests that p53 recruitment to the BLM promoter is dependent on p130 (Figure S5B).

8. One of the intriguing findings of this study is that when either of the two most common p53 mutants is overexpressed, they are now able to interact with E2F4 (unlike the WT p53). These mutants are not recruited to the BLM promoter and, moreover, their expression results in the major reduction in DREAM binding to the BLM promoter and, consequently, in defective DREAM-mediated transcriptional repression of relevant promoters. Both p53 mutants have previously been shown to spuriously bind to and cause aggregation of other proteins, e.g. p63 and p73. It is therefore possible that these p53 mutants exert similar activity towards E2F4. Based on all the data presented and described in Fig.4 and S4, the title of the section "E2F4 sequesters p53 cancer mutants" should probably be altered to something like "p53 cancer mutants sequester E2F4".

Response: We agree with the reviewer's comment. We have modified the above statement as "p53 cancer mutants sequester E2F4" (Page 9).

9. The interpretation that the "non-canonical" function of p53 is to recruit DREAM complex to the target promoters is not rigorously established. If those promoters do not contain p53 consensus binding sites then p53 is recruited there indirectly, i.e. via its interaction with DREAM complex. This is indeed what has been observed in this study. The Re-ChIP in Fig.5 and S5, is also done in a way of performing E2F4 ChIP first followed by p53 ChIP. Thus, a model in which the DREAM complex recruits p53 to (some) of its promoters is probably more accurate. The role of p53 might be in stabilising the DREAM complex at promoters rather than in recruiting it.

Response: We appreciate the reviewer's perspective and fully agree with the comments. Accordingly, we have addressed this in the revised manuscript by including a statement in the Discussion section, stating that. "This direct repression by p53 via the "non-canonical" pathway is achieved when the DREAM complex recruits p53 to some of its target promoters in a p21-independent manner (Figure 6). We provide evidence that this mechanism occurs because p53 interacts with the two most critical DREAM complex components, p107/p130, irrespective of their phosphorylation status. The recruitment of p53 to the DREAM complex possibly enhances the repertoire of promoter selections and may also increase stability of this complex." (Page 12).

Reviewer #2

One key mechanism by which the tumor suppressor p53 represses transcription has been proposed to involve the DREAM complex. Upregulation of the cyclin-dependent kinase inhibitor p21 by p53, in turn, causes dephosphorylation of key DREAM components, leading to suppressed gene expression. In this intriguing study, the authors present data to argue that p53 can repress via the DREAM complex in a manner independent of p21. And that this repression occurs with a transactivation-deficient mutant of p53. They propose a "non-canonical" pathway for DREAM-mediated repression that involves a direct interaction of the complex with p53.

Understanding mechanisms by which p53 regulates gene expression is a critically important area given the relevance of p53 as a tumor suppressor in human cancer. The notion that p53-dependent repression occurs in a p21-independent manner via the DREAM complex is a new and exciting finding. Thus, the manuscript has both high significance and novelty making it suitable for the readership of The EMBO Journal. However, the data as presented is too preliminary at this time and lacks essential controls experiments.

Response: Response: We thank the reviewer for the positive comments. In the revised version, we have tried to answer almost all the reviewers' queries.

The manuscript needs substantial revision before it is ready for publication. Key issues are as follows:

First, a key finding in the study is the p21-independence of the effects. The authors show data using the HCT116 cell line that has been engineered by homologous recombination to no longer express 21. These findings need to be validated in other cell lines using RNAi or other means to knockdown p21 expression to confirm that effects are not peculiar to a specific cell line.

Response: We appreciate the reviewer's comment.

We have performed the key assays in the HT1080 isogenic cells (WT, p53^{-/-} and p21^{-/-} genotypes) and U2OS, where p53 and p21 have been ablated.

A. In U2OS cells, where p53 and p21 were ablated, RT-qPCR and western blot analyses revealed that similar to HCT116 cells, the levels of DREAM complex targets at both transcript and protein increased more significantly in the absence of p53 compared to the absence of p21 (Figure 2E, 2F).

B. In HT1080 isogenic cells, protein levels of DREAM complex targets are determined through western (Figure 2D), DNA affinity experiment on BLM promoter (Figure 3D) and E2F4-p53 Re-ChIP carried out and similar results were obtained as HCT116 cells (Figure S5A).

Second, the use of the doxycycline-inducible expression of the transactivation-deficient p53 mutant is central to one of the key conclusions. It is unclear whether the levels of p53 that is achieved in this engineered system have biological relevance. Levels should be compared to cells with endogenous p53 expression to ensure that the expression is not artifactually high. This would be one explanation for the surprising finding that the transactivation-dead mutant retains activity in these assays.

Response: We agree with the reviewer's comment. We have compared the levels of p53 in HCT116 WT, HCT116 p53^{-/-} p53 WT and HCT116 p53^{-/-} p53 TAD mutant cells. The levels of p53 are only around 1.25 fold more than the endogenous levels in HCT116 WT cells (Figure S1G).

Third, the ChIP-seq experiments need to be performed in the p21-null cells. An important finding is that the DREAM repression is p21-independent. The authors propose a mechanism to explain this involving p53 recruitment to genes by DREAM in the absence of p21. This needs to be directly demonstrated experimentally.

Response: We appreciate the reviewer's comment. We have conducted E2F4-p53 Re-ChIP seq in HCT116 p21^{-/-} as well as in HCT116 p53^{-/-} expressing p53 TAD mutant (in the presence of Doxycycline). The results show that even in cells lacking p21, p53 can be recruited to the DREAM complex target promoters. We have incorporated the results in the manuscript (Figure 5B, 5C, 5E, 5G, S7A-S7C).

Fourth, many of the conclusions are supported by immunoblotting data. Since it is to be presumed that these blots were replicated, quantitation of multiple blots accompanied by some statistical analysis would strengthen conclusions, rather than merely showing a single replicate blot.

Response: We agree with the reviewer's comment. We have quantified all the rounds of immunoblotting data and presented the data as an Excel sheet in Supplementary Information Table S4. The values showing the quantitation of individual blots have been removed from the figures.

Fifth, much of the published literature has addressed the role of DREAM in the response of p53 to DNA damage. It is unclear how to integrate the findings shown here with what has been previously shown. This is important as the authors are proposing a "non-canonical" pathway. One possibility is that the p21-dependence of DREAM effects may be different basally versus in the DNA damage response. This needs to be addressed so that the current findings can be clearly placed in the context of what has previously been published.

Response: We appreciate the reviewer's comment. We treated HCT116 isogenic cells with DNA-damaging agents, including 5-FU, Doxorubicin, and Nutlin. Our results showed a reduction in DREAM target levels in HCT116 p53^{+/+} and HCT116 p21^{-/-} cells in response to DNA damage. However, no such effect was observed on DREAM targets in HCT116 p53^{-/-} cells (Figure S2A–S2C).

Sixth, given the model being proposed, it is confusing why occupancy by p53 is not revealed in p53 ChIPseq experiments. Although the model suggests that p53 is recruited by DREAM, one would expect p53 occupancy still to be detected by a direct ChIP analysis using a suitable p53 antibody.

Response: We appreciate the reviewer's comment. We have performed p53 ChIP-qPCR on the BLM promoter using C-terminus, Pab421 p53 antibody p53 under asynchronous conditions, revealing basal-level recruitment of p53 to the #2 site on the BLM promoter. To further confirm p53 recruitment, cells were treated with DNA damaging agent 5-FU, which led to enhanced p53 recruitment at the #2 site of the BLM promoter in both HCT116 WT and HCT116 p21^{-/-} cells (Figure S3J).

Seventh the authors focus on four targets, BLM, RAD54, RAD51, and BRCA1 to show p21-independence. They need to show other targets that are p21-dependent and address what may be possible determinants for which pathway is utilized for repression. Is this target gene-specific? And if so, is there a sense of why? It is unclear whether the findings in Figure 5 reflect the sensitivity of assays or that there are in fact distinct targets for p53 association.

Response: We appreciate the reviewer's comment and concerns.

A. To address the p21 independent repression on other DREAM complex targets, E2F4-p53 Re-ChIP seq was performed in HCT116 p21^{-/-} and HCT116 p53^{-/-} p53 TAD mutant cells in the presence of Doxycycline. Re-ChIP seq data showed E2F4-p53 co-recruited to BLM, RAD54, as well as BUB1, MELK (Figure 5G, S7A-S7C) (other known DREAM targets) and including 399 other targets (which also includes canonically regulated DREAM complex targets) commonly in all three cells i.e p53^{+/+}, p21^{-/-} and p53 TAD mutant cells (Figure 5F).

B. Furthermore, comparing only p21^{-/-} and p53 TAD mutant cells, E2F4 and p53 were found to be co-recruited to 2723 additional targets that are not targeted by p53 WT. This phenomenon possibly indicates that the recruitment of transactivation-deficient p53 may serve as a compensatory mechanism via which the cells were trying to prevent the initiation or propagation of genomic instability which results in vivo due to the loss of p21 or transactivation function of p53. Altogether the meta-analysis of the ChIP-seq experiments highlights the dynamic nature of p53's role in maintaining genomic integrity, even under conditions of impaired “canonical pathway” (Page 13-14)

Eighth, additional mechanistic insight is needed to explain the consequence of p53 in the complex that is assembled on specific promoters. The presence of the transactivation-dead mutant in this complex and how this generates a repression signal also needs to be addressed. Why does the presence of p53 at these promoters in the context of DREAM result in repression? What is happening to the potent activation domain of p53 in this complex? The model presented in Figure 6 needs clarification. It is unclear what is the difference between the canonical and non-canonical pathways. What does the presence of p53 at the promoter add?

Response: We appreciate the reviewer's comment and concern, in response eighth point has been further sub-divided into four sections:

A. Additional mechanistic insight is needed to explain the consequence of p53 in the complex that is assembled on specific promoters.

The recruitment of p53 to the DREAM complex under asynchronous conditions may facilitate DREAM complex formation by its interaction with the spacer domain of p130/p107

(Figure 3E-3J, S4G-S4L). This interaction likely contributes to stabilizing the DREAM complex, regardless of the phosphorylation state of p130/p107.

B. The presence of the transactivation-dead mutant in this complex and how this generates a repression signal also needs to be addressed. Why does the presence of p53 at these promoters in the context of DREAM result in repression?

The presence of a trans-activation dead mutant (TAD) was demonstrated genome-wide through E2F4-p53 Re-ChIP sequencing in HCT116 p53^{-/-} cells expressing the p53 TAD mutant (Figure 5C, 5F, S7A-S7C). Under asynchronous conditions, in the presence of WT p53 or the p53 TAD mutant, p130/p107 remains hyperphosphorylated. However, we have shown that both p53 and the p53 TAD mutant interact with p130/p107 via their DNA-binding domain, regardless of the phosphorylation state of p130/p107 (3E-3J, S4G-S4L). This recruitment of p53 or the p53 TAD mutant to the DREAM complex likely contributes to the formation and stabilizing of this complex.

C. What is happening to the potent activation domain of p53 in this complex?

To date, the canonical pathway of DREAM complex-mediated repression is known to function primarily in response to DNA damage or during cellular senescence (G0 phase). In both scenarios, either active p53 or p53-mediated activation of p21 is essential, in such conditions the activation domain of p53 plays a critical role.

In contrast, the non-canonical pathway enables DREAM complex-mediated repression under asynchronous conditions, independent of both p53's activation domain and p21. This indicates that the activation domain of p53 is not required for this pathway, as demonstrated in cells expressing the p53 TAD mutant.

D. The model presented in Figure 6 needs clarification. It is unclear what is the difference between the canonical and non-canonical pathways. What does the presence of p53 at the promoter add?

We have updated Figure 6 to illustrate that when p53 is activated (in response to stress or damage), it triggers the activation of p21. This, in turn, promotes the hypophosphorylation of p130/p107, facilitating the formation of the DREAM complex via the canonical pathway.

In contrast, under asynchronous conditions, when WT p53 or the p53 TAD mutant does not trans-activate p21, p130/p107 remains hyperphosphorylated. Our findings demonstrate that this hyperphosphorylated form can still interact with WT p53 or the p53 TAD mutant, resulting in the formation of a stable DREAM complex through the non-canonical pathway.

Additional comment:

1. In the Introduction, the authors state unequivocally that previous studies show that p53 represses indirectly via DREAM. There is a substantial published literature that supports the idea that, at least on some targets, p53 likely acts directly to repress.

Response: We appreciate the reviewer's comment. We have modified the introduction, stating that "The role of p53 as a repressor has been re-evaluated in the recent past. p53 has been known to directly repress genes by binding to its response elements and recruiting corepressors or by blocking transactivators from accessing their binding sites (Brady & Attardi, 2010; Ho & Benchimol, 2003)." (Page 3).

Dear Dr. Sengupta,

Thank you for submitting a revised version of your manuscript. Your study has now been seen by all original referees, who find that their previous concerns have been addressed and now recommend publication of the manuscript. There remain only a few mainly editorial points that have to be addressed before I can extend formal acceptance of the manuscript:

- Please remove the figures from ms file and leave them only as individual, high-resolution Figure files, no track changes
- Please add missing FUNDING INFO in eJP: Indo- French Centre for Promotion of Advanced Research (IFCPAR/CEFIPRA) grant IFC/6803- 1/2022 and Council of Scientific and Industrial Research (CSIR), India grant 27/0387/23/EMR-II
- Please rename the Conflict of Interest section into "Disclosure and Competing Interests Statement", in accordance with our updated Guide to Authors (<https://www.embopress.org/competing-interests>)
- Please update source file names, titles, legends and manuscript callouts to Dataset EV1-EV# instead of Tables S1, S2 and S4
- APPENDIX 1 FILE WITH ToC: "Appendix for..." should be added before the ms title on the title page, missing ToC with page numbers of the listed items; nomenclature should be Appendix Figure Sx throughout the Appendix PDF and ms file; Table S3 should be renamed to Appendix Table S1 throughout the Appendix PDF and ms file
- R&T TABLE: in ms file, needs to be uploaded as individual file using the template from GTA
- Please save the Source data files in a scheme one figure/folder and then upload as .zip files. E.g. all the Source data files for figure 1 need to be saved in a single folder and this needs to be zipped and then uploaded as "SD figure 1.zip" file. For EV and/or appendix figures, ZIP together all source data. SD checklist should be uploaded as Related Manuscript File.
- Figure Legends (main + EV):
 1. Please note that the exact p values are not provided in the legends of figures 1A, F; 2E, G, H, J, K; 3K, 4B, F, G.
 2. Please indicate the statistical test used for data analysis in the legends of figures 1A, F; 2A, E, G, H, I, J, K; 3K, L; 4B, F, G; 5H.
 3. Please note that the error bars are not defined in the legends of figures 2E.
- Please adjust the order of the manuscript sections: Title page with complete author information, Abstract, Keywords, Introduction, Results, Discussion, Methods, Data Availability Section, Acknowledgements, Disclosure and Competing Interests Statement, References, Main figure legends, Tables, Expanded Figure Legends.

With best regards,

Cornelius Schneider

Cornelius Schneider, PhD
Editor
The EMBO Journal
c.schneider@embojournal.org

We realize that it is difficult to revise to a specific deadline. In the interest of protecting the conceptual advance provided by the work, we recommend a revision within 3 months (13th Apr 2025). Please discuss the revision progress ahead of this time with

the editor if you require more time to complete the revisions. Use the link below to submit your revision:

Referee #1:

The data are of good quality and the conclusions reached appear to me to be valid and correct.

I am pleased to write that the authors have done a good job of successfully addressing my suggestions and requests. In light of this, I feel that the paper should be accepted for publication in EMBO J.

Referee #2:

The authors have adequately addressed the concerns of the previous review. The manuscript is now acceptable for publication in EMBO J.

All editorial and formatting issues were resolved by the authors.

Dear Dr. Sengupta,

I am pleased to inform you that your manuscript has been accepted for publication in the EMBO Journal.

Yours sincerely,

Cornelius Schneider, PhD
Editor
The EMBO Journal
c.schneider@embojournal.org
